# Improving Generalization in ML models via Causal Interaction Constraints

## Abstract

Machine learning models are effective in identifying patterns within independently and identically distributed (i.i.d.) data. However, this assumption rarely holds in real-world applications, where violations of i.i.d. can hinder both generalization and explainability. Causal Machine Learning is an emerging discipline that addresses these limitations by integrating causal reasoning, an element typically absent from conventional approaches. In this work, we introduce a novel causal machine learning strategy that emphasizes the role of spurious variable interactions, a concept grounded in the Independent Causal Mechanisms (ICM) principle. We argue that recognizing and constraining these spurious interactions is essential for improving model robustness and interpretability. To that end, we introduce a novel approach for incorporating interaction restrictions into neural network architectures and tree-based models such as random forest and gradient boosting. When applied to real-world scenarios, our method demonstrates that predictive models explicitly constrained to avoid spurious interactions exhibit enhanced generalization performance across diverse domains, outperforming their unconstrained counterparts.

## 1 Introduction

Machine learning techniques have achieved impressive results in pattern recognition and prediction tasks under the assumption that data are independently and identically distributed (i.i.d.). However, in real-world applications, this assumption often breaks down, posing significant challenges for model generalization. A key limitation lies in the fact that conventional machine learning models typically do not incorporate causal knowledge, restricting their ability to adapt across diverse environments or respond reliably to interventions (Peters et al., 2017; Goyal & Bengio, 2022; Pearl, 2019; Ahmed et al., 2020). To bridge this gap, there has been a growing interest in the field of Causal Machine Learning, which aims to infer causal relationships from data and enhance conventional machine learning techniques by integrating causal knowledge that is typically overlooked by standard approaches (Kaddour et al., 2022; Binkyte et al., 2025).

Understanding real-world problems requires recognizing that they are shaped by underlying data-generating processes. These processes can often be captured through hierarchical representations, as highlighted by Schölkopf et al. (2021) in the context of causal representation. In their work, the authors note that the gold standard for modeling natural phenomena is typically a system of coupled differential equations, which describe the physical mechanisms driving a system's evolution over time. Such models enable the prediction of future behavior, the assessment of intervention impacts, and the identification of statistical dependencies among variables.

While differential equations offer a comprehensive description of systems and their causal structures, statistical models often provide a more limited understanding. They generally focus on how certain variables can predict others under fixed experimental conditions, without referencing the underlying dynamic processes. Statistical and machine learning models are valuable for learning from i.i.d. data but often fall short when it comes to predicting the effects of interventions. Causal modeling offers a middle ground between these approaches, aiming to generate insights and predict intervention outcomes through data-driven methods. It

thereby substitutes some of the expert knowledge required for differential equations with weaker and more generic assumptions.

To generalize effectively beyond their training data, machine learning models must capture key properties of the underlying data-generating process. This is made possible through inductive biases, built-in assumptions that steer the learning process toward more plausible and generalizable solutions (Baxter, 2000). By narrowing the hypothesis space, inductive biases help prevent overfitting and enhance model robustness. Examples include regularization methods, model selection strategies, parameter constraints, and architectural choices in neural networks.

Grounded in causal inference theory, the Independent Causal Mechanisms (ICM) principle (Peters et al., 2017) states that the generative process of a system's variables consists of autonomous modules, each responsible for a distinct part of the process. These modules operate independently, meaning that the conditional distribution of each variable, given its direct causes, remains stable even when other mechanisms in the system are altered. This principle captures core ideas in causal reasoning, including modularity, subsystem autonomy, and the independent intervenability of causal variables. When applied to causal factorization, the ICM principle implies that the different components of a system should be independent in two key ways. First, if we intervene on one mechanism, changing how a particular variable depends on its direct causes, it should not affect how any of the other variables are generated. Second, knowing how some parts of the system work should not give us any information about how other parts work; each mechanism operates independently of the others.

Machine learning models that do not account for these independences tend to deviate from the true data-generating process, leading to a lack of robustness against interventions and data shifts (Schölkopf et al., 2021). While such models may perform well on training data, they frequently fail to generalize to new, unseen scenarios where parts of the underlying generative mechanisms have changed (Subbaswamy et al., 2022).

In this work, we introduce a new approach to improving out-of-distribution robustness by incorporating the inductive bias implied by the Independent Causal Mechanism (ICM) assumption directly into predictive modeling. Our first contribution is to show that constraining interactions according to ICM, an idea previously applied mainly in causal inference (Pros & Vitrià, 2025; 2023), can also play a central role in a different context: robustness to distributional shift. We provide extensive examples illustrating its implications across different types of shifts. Second, we propose a new modification of the masking mechanism from Parafita & Vitrià (2022) that allows this bias to be encoded efficiently in neural networks. For completeness, we also describe how the same inductive bias can be integrated into tree-based models, even though we make no algorithmic contribution in that setting. At a high level, our method works by enforcing structured interaction constraints that reflect the independencies of the true data-generating process, leading to models that generalize more reliably, remain stable under interventions, and offer improved interpretability.

In Section 2, we highlight the role of inductive biases in developing robust models and interpret distribution shifts as interventions within a causal framework. Section 3 introduces our approach to interaction selection, emphasizing the need to prevent spurious interactions that deviate from the true data-generating process. Section 4 details our methodology for incorporating causal inductive biases into both tree-based and neural network models. In Section 5, we present experimental results on synthetic and real-world datasets, demonstrating the effectiveness of our approach in improving out-of-distribution performance. Finally, Section 6 concludes with a discussion of our findings and their implications for future research in causal machine learning.

## 2   Inductive biases for robust models

Our first objective is to prevent predictive models from relying on variable associations that are not robust under distribution shifts. Such spurious associations may hold in the training data but fail to persist when the underlying data distribution changes, leading models to make unreliable or misleading predictions. This vulnerability significantly limits the model's applicability in real-world settings, where conditions often differ from those seen during training due to interventions, environmental changes, or domain shifts. Ensuring

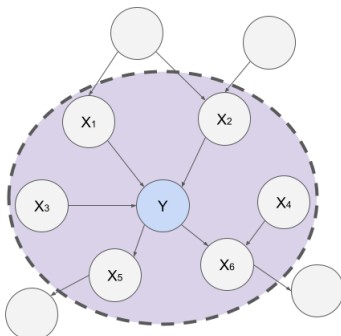

Figure 1: The Markov boundary of a target variable $Y$ (shown in blue) is the minimal set of variables $\{X_1, X_2, X_3, X_4, X_5, X_6\}$ (shaded region) that renders all other variables conditionally independent of $Y$. This set includes the direct causes, direct effects, and the direct causes of the direct effects of $Y$. Variables outside the boundary (shown in gray) do not provide additional information about $Y$ once the Markov boundary is known. Identifying the Markov boundary is crucial for tasks such as feature selection, causal discovery, and interpretability in machine learning.

robustness to these shifts is therefore essential for building models that maintain their performance across diverse contexts. To achieve this, it is important to identify and prioritize stable, causally grounded relationships over purely statistical associations, thereby aligning model behavior more closely with the underlying data-generating mechanisms.

A data or distribution shift can be interpreted as a form of intervention on the data-generating process (Subbaswamy et al., 2022). More broadly, dataset shifts refer to discrepancies between the environment in which a model is trained and the one in which it is ultimately deployed. These shifts often manifest as changes in the statistical properties of the data, such as feature distributions, causal mechanisms, or outcome-generating processes, and can arise from various real-world factors, including changes in population demographics, sensor noise, policy interventions, or external events.

By viewing such shifts through the lens of causal inference, we gain a principled framework for analyzing and addressing them. Specifically, understanding distribution shifts as interventions allows us to reason about how and why the data distribution changes, and to distinguish between stable (invariant) and unstable (intervention-sensitive) components of the model. This causal perspective provides a foundation for developing methods that enhance model robustness by focusing on features and relationships that remain consistent across environments. Ultimately, incorporating causal reasoning into the treatment of distribution shifts helps bridge the gap between training conditions and deployment scenarios, leading to models that generalize more reliably in dynamic, real-world settings.

## 2.1 Causal Machine Learning

Causal approaches for learning models that are robust and transferable across domains can be broadly categorized into two main classes (Kaddour et al., 2022). The first, **invariant feature learning** (Arjovsky et al., 2020), focuses on identifying a set of features $C$ such that the conditional distribution $P(Y|C)$ remains stable across different environments. The goal is to find representations that preserve the predictive relationship to the target variable $Y$ regardless of shifts in the input distribution. The second class, **invariant mechanism learning** (Schölkopf et al., 2021), seeks to uncover the underlying data-generating mechanisms that define different interventional distributions. This approach emphasizes modeling the structure of the causal process itself, enabling the model to adapt to changes that arise from interventions or shifts in the environment.

Invariant feature learning is a type of variable selection. The task is to identify features of the data that are predictive of the outcome in distinct environments. Variables closer to the target are considered better predictors, leading to causal feature selection approaches. For example, from a causal mechanisms perspective,

the causal parents of the outcome are always predictive except for a shift in the distribution of the outcome itself.

In general, causality-based feature selection methods aim to obtain the Markov boundary of the target variable (see Figure 1) for achieving explainable and robust machine learning methods (Yu et al., 2020). The Markov boundary of the target variable is the minimal feature subset with maximum predictivity since all other features are probabilistically independent of the target variable conditioning on its Markov boundary (Pearl, 2014; Koller & Sahami, 1996; Tsamardinos & Aliferis, 2003). Other approaches include focusing on learning a sufficient subset of invariant features (Kim et al., 2025) or using causal inference to detect causal variables unaffected by the environment (Su & Wang, 2024).

Invariant Mechanism Learning (IML) goes beyond invariant feature learning by aiming to uncover a collection of data-generating mechanisms that remain stable across different interventional distributions. Grounded in the principle of Independent Mechanisms, IML seeks to disentangle what is being conveyed from how it is conveyed. This is similar to how humans can understand spoken content regardless of whether it is delivered in a loud or quiet voice, we separate the message from the mode of delivery. In the same way, IML attempts to isolate the underlying causal mechanisms from the varying contexts or environments in which they operate. IML seeks to model interventions on subsets of these confounders to improve robustness in predictions. Notably, Yue et al. (2021) suggest framing unsupervised domain adaptation for classification as the task of uncovering a set of disentangled causal mechanisms that map the source domain to the target domain. In a different vein, methods such as those proposed by Cui & Athey (2022) leverage techniques like reweighting and counterfactual reasoning to introduce causal biases that enhance model robustness and interpretability. These strategies reflect a broader trend toward integrating causal insights into machine learning to improve generalization across environments.

In this work, we focus on constraining the functional form of predictive models by explicitly accounting for interactions between variables. To better encode the set of independencies implied by the ICM assumption, the model must adhere to specific constraints that reflect the structure of the true data-generating process. In Section 3, we formalize and illustrate these constraints, both in the general case and in scenarios where the nature of the interventions is known. This allows us to design models that are not only robust to distribution shifts but also more interpretable.

## 2.2 ML and robustness to interventions

In certain settings, the nature of the intervention inducing a distributional shift is known a priori. For instance, in medical imaging, a predictive model trained on data acquired from one imaging device may be applied to data from a different device, resulting in systematic discrepancies in resolution, contrast, or noise characteristics. Such covariate shifts can introduce distributional biases that compromise model performance if not properly accounted for. When the intervention is identifiable, it is possible to design models that are intrinsically robust to these shifts by explicitly modeling and incorporating the underlying changes in the data-generating process.

Kaddour et al. (2022) propose an approach that incorporates information from the causal graph as a form of regularization when interventions affect non-causal features. In contrast, other methods focus on interventions within the causal structure itself. For example, Makar et al. (2022) introduce a causally motivated regularization technique that uses auxiliary labels to discourage models from relying on shortcut features, thereby enhancing robustness under distribution shifts caused by known interventions. Similarly, Subbaswamy et al. (2022) present a unifying causal framework that integrates various strategies for building robust predictive models in the presence of known interventions. These approaches are founded on the notion of stable distributions, which serve as target distributions for mitigating the effects of instability induced by distributional shifts.

This framework introduced by Subbaswamy et al. (2022) further introduces a causal hierarchy of stable distributions, offering theoretical guarantees regarding the stability and generalizability of predictive models across diverse environments. The first level of their framework (Level 1) focuses on **observational conditional distributions**, addressing the challenge of feature selection under known interventions. This is achieved by, given a set of features $X$, identifying a subset $Z \subseteq X$ such that the conditional distribution

$P(Y \mid Z)$ remains invariant across environments. The selection process excludes variables associated with unstable edges in the causal graph, ensuring that $P(Y \mid Z)$ is robust to changes elsewhere in the system. While this strategy may omit stable yet predictive pathways, potentially impacting model accuracy, it offers strong robustness guarantees. Importantly, it operates solely on observational data, making it particularly valuable in settings where interventional or counterfactual information is not accessible.

The second level uses conditional interventional distributions (Level 2), employing interventional distributions of the form $P(Y \mid \mathrm{do}(W), Z), W \in X$, where the do-operator removes the influence of unstable mechanisms by graphically "cutting" edges into intervened variables. Unlike Level 1, this method retains stable dependencies by intervening on variables $W$ whose mechanisms are subject to shifts. This method is particularly useful when domain knowledge identifies specific mutable mechanisms (e.g., hospital-specific protocols) but requires identifiability conditions (e.g., no unmeasured confounding of $W$).

## 3 Interaction Selection

To create robust and explainable predictive models, it is necessary to approximate the distribution of the data generating process, as only associations arising from causal relationships reflect the intrinsic dependencies between variables (Cui & Athey, 2022). Other types of associations are spurious and depend on the joint distribution of features and data collection processes, making them sensitive to changes in these factors and fail to preserve the independence structure inherent to Independent Causal Mechanisms (ICMs).

In causal inference, the term **mechanism** describes the underlying processes or relationships that determine how a certain variable depends on its direct causes. Each mechanism captures a stable, autonomous relationship, meaning it remains consistent even if other parts of the system change.

Causal graphs provide a simplified visual representation of these relationships. In these graphs, each node represents a variable, and arrows indicate direct causal influences. However, a single node can represent multiple distinct mechanisms because the mathematical relationship (or structural equation) between a variable and its direct causes might encode separate, independent causal pathways.

For instance, the mechanical energy of an object is the sum of its kinetic and potential energy, given by $E = \frac{1}{2}mv^2 + mgh$, where the mass $m$ influences energy through both velocity $v$ and height $h$ independently. Although $\{m, v, h\}$ are all direct causes of $E$ in the causal graph, the structure of the equation reveals two distinct mechanisms: one involving $\{E, m, v\}$ and another involving $\{E, m, g, h\}$. Both mechanisms operate independently. Changing the height $h$ does not affect how mass $m$ and velocity $v$ produce kinetic energy, and similarly, altering velocity $v$ does not impact the way height $h$, gravity $g$, and mass $m$ determine potential energy.

While causal graphs help visualize direct causal relationships clearly, they alone don't always explicitly represent these independent mechanisms. To fully and explicitly capture these independent causal mechanisms, researchers use Structural Causal Models (SCMs) (Pearl, 2009). SCMs extend causal graphs by specifying the exact mathematical form of the relationships, making explicit the independence between different causal processes.

In this context, distribution shifts can arise under two primary conditions: (1) when the domain of a variable changes, for example, due to sampling bias or external interventions that modify its observed range; or (2) when there is a change in the underlying causal mechanisms themselves, for instance through shifts in parameters or structural alterations in the data-generating process. The former scenario impacts the distribution of observed variables without modifying their intrinsic functional relationships, whereas the latter scenario directly alters the mechanisms that define interactions among variables, resulting in more substantial and fundamental changes in predictive behavior.

To identify stable distributions that accurately approximate the causal relationships inherent in the data-generating process, we argue that it is essential to explicitly consider interactions among variables. Formally, a function $F(\mathbf{x})$ is said to exhibit an interaction between two variables $x_j$ and $x_k$ if the change in the value of $F(\mathbf{x})$ resulting from modifying the value of $x_j$ depends explicitly on the value taken by $x_k$. For numeric

variables, this interaction is expressed as

$$\frac{\partial^2 F}{\partial x_j \partial x_k} \neq 0 \tag{1}$$

or by an analogous expression for categorical variables including finite differences.

As an illustrative example, consider the relationship between physical exercise and weight change. One might hypothesize that an increase in daily exercise duration leads to a reduction in body weight. However, the magnitude of this effect may depend on an individual's daily caloric intake. For individuals with relatively low caloric intake, increasing exercise from 30 to 60 minutes per day may result in substantial weight loss. In contrast, for individuals with high caloric intake, the same increase in exercise duration may yield minimal or negligible weight change, as the additional calories consumed may offset the energy expended through exercise.

In this context, we aim to develop a predictive model to estimate weight change based on daily exercise and calorie intake. An interaction between these variables implies that the effect of one predictor (e.g., exercise) on weight change depends on the level of the other (e.g., caloric intake). Accounting for such interactions is crucial to enhance both the accuracy and the interpretability of the model.

In the same context, consider the variables that represent exercise duration and age. Suppose we find that increasing daily exercise, for example, from 30 to 60 minutes, leads to a similar amount of weight loss regardless of a person's age. Likewise, any influence that age might have on weight change (such as a slower metabolism with aging) occurs independently of how much the person exercises. In this case, the two variables, exercise and age, do not interact, because the effect of one does not depend on the level of the other. Including this interaction term in a predictive model allows the effect of one variable to depend on the level of another, capturing more complex relationships in the data. This makes interpretation more nuanced. It also increases model complexity and can lead to overfitting if the interaction is not truly supported by the data.

Even a highly accurate predictive model can contain substantial interaction effects that are not supported by the data generation process. These spurious interactions can occur when there is a high degree of collinearity among some (or all) of the predictor variables in the training data. Since the data-generating process does not contain such interactions, predictions based on them will not be robust to interventions and thus to distribution shifts.

---

**Algorithm 1** Variable Interaction Selection

---

1: **Input:** Set of variables $X$, causal knowledge (graph or expert knowledge)
2: Initialize empty collection $\mathcal{G} \leftarrow \emptyset$
3: **for** each known stable ICM $M$ **do**
4:      Initialize empty group $G \leftarrow \emptyset$
5:      **for** each variable $x \in X$ **do**
6:          **if** $x$ belongs to mechanism $M$ **then**
7:              Add $x$ to $G$
8:          **end if**
9:      **end for**
10:      Add group $G$ to $\mathcal{G}$
11: **end for**
12: **Output:** Collection $\mathcal{G}$ of variable subsets, where interactions are allowed only within each subset

---

For instance, in Example 1, directly modelling $P(Y|A, B)$ could include interactions between $A$ and $B$ which are not present in any mechanism when factorized according to the causal graph assumption $P(A, Y, B) = P(A)P(Y|A)P(B|Y)$. From a causal perspective, the parents of a target variable $Y$ remain predictive of $Y$ under any interventional distribution, except when $Y$ itself is directly intervened upon. While incorporating additional variable interactions can enhance model performance, it also makes the model vulnerable to interventions affecting other parts of the data generating process.

Causal assumptions, expressed as partial expert knowledge or a complete causal graph, can provide information about the nature of the interactions between variables in the data-generating process. For example, consider the Markov blanket in Figure 1. Variables $X_1$ and $X_4$ correspond to distinct mechanisms, so using an interaction between these two variables to predict the target $Y$ would not be robust to domain shift. In algorithm 1 we detail the interaction selection process.

Alternatively, by carefully selecting interactions based on causal relationships, we can enhance the explainability and robustness of predictive models to distribution shifts, improving the reliability of their predictions even when the underlying data distribution changes. In Example 1 we use a simple case to illustrate the impact of using an interaction not present in the real model when the model is used for out-of-distribution prediction.

---

### Example 1

**Description**

The dataset simulates the relationship between three variables: attending a coding bootcamp ($A$), programming skills ($Y$), and landing a high-paying tech job ($B$). Attending a coding bootcamp ($A$) directly contributes to developing programming skills ($Y$) through focused and intensive training. These programming skills translate the education and experience gained at the bootcamp into tangible qualifications, which are critical for competing in the tech industry. Ultimately, the acquisition of programming skills enables the outcome of landing a high-paying tech job ($B$). This setting implies that the interaction between A and B is spurious. In the absence of intervention, the best predictive model is an unrestricted one that includes both $A$ and $B$, capturing their observed relationship. However, under intervention such as the change of industry, where the relationship between $Y$ and $B$ is modified, a restricted model that prevents this interaction performs better. However, in such cases, a causal feature selection strategy such as the Level 1 methods described in Section 3, which rely on a model using only $A$, yields the most accurate results by isolating the direct effect of the intervention.

**Data Generating Process**

$$\beta = 3$$
$$A \sim \mathcal{N}(0,1)$$
$$Y = \beta A + \mathcal{N}(0,1)$$
$$B = 2Y + \mathcal{N}(0,4)$$
$$\beta_{int} \sim \mathcal{N}(0,2)$$
$$Y_{int} = \beta_{int} A + \mathcal{N}(0,1)$$

**Known mechanisms**

$$Y = f_Y(A, \epsilon_Y)$$
$$B = f_B(Y, \epsilon_B)$$

Figure 2: Causal graph for example 1.

**Results**

| Model | Test MSE (SD) | Intervention MSE (SD) |
|---|---|---|
| Unconstrained | 1.111 (0.171) | 11.421 (10.204) |
| Interaction Constraints | 1.113 (0.180) | 7.675 (6.598) |
| Stable Features | 4.385 (0.689) | 3.904 (0.819) |

Table 1: Test and intervention MSE results for different models. Unconstrained is a regular predictive model using both A and B features, Interaction Constraintsuses the approach described in section 4.1 to remove the interaction between A and B and Stable Features model uses only A.

---

In addition to enhancing out-of-distribution performance, developing a model that more closely aligns with the true data-generating process can improve explainability, as variable contributions become more strongly correlated with true causal variable contributions. Figure 3 illustrates the correlation between SHAP values for models with and without interaction terms and the true causal model as noise levels increase, in the

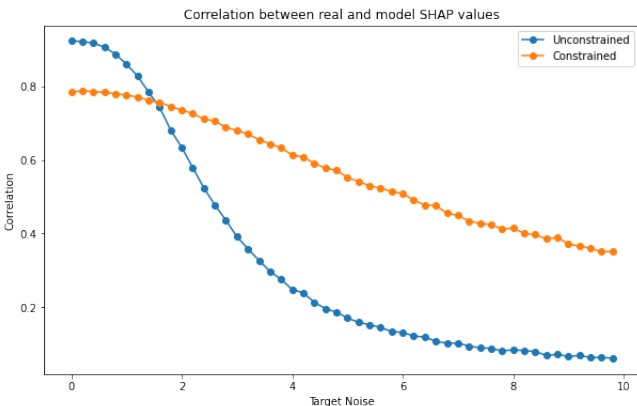

Figure 3: Correlation between true Data Generating Process SHAP values and model SHAP values for ICM-constrained and unconstrained models under varying noise scenarios. The constrained model maintains higher correlation as noise increases. Results averaged over 100 runs (max standard deviation < 0.01).

scenario described in Example 1. While the unconstrained model accurately captures the causal model under low noise conditions, its correlation with the true causal model diminishes as noise increases. This issue can be mitigated by incorporating interaction selection into the model, thereby maintaining closer alignment with the true causal model even under higher noise levels.

So far, we have treated interventions as unknown distribution shifts that may occur after model training. However, in practice, certain interventions can be anticipated, such as policy changes, experimental conditions, or domain-specific adjustments. When the nature of an intervention is known in advance, for example the unstable mechanisms in Example 2 can be identified in the term $P(O|S, D)$, specialized methods can be employed to explicitly account for its effects during model development.

**Interaction selection under known intervention**

When the intervention is known, we can leverage the methodological approaches outlined in Section 2.2. These methods provide a structured framework for incorporating prior knowledge of the intervention into the model, thereby improving inference and prediction accuracy. By explicitly modeling the known intervention, we can reduce uncertainty and enhance the interpretability of results.

Additionally, the methods introduced in Section 3 are compatible with the interaction selection approach described in the previous section. This compatibility allows for a unified modeling framework where both intervention knowledge and interaction effects are simultaneously accounted for. By integrating these two approaches, we can refine our analysis to better capture the underlying causal mechanisms and dependencies within the data.

To illustrate the practical advantages of this combined approach, we consider Example 2, which presents a case where both known intervention methods and interaction selection play a crucial role. In this example, the intervention introduces structured changes in the system, and interaction effects further modulate the outcome. By applying the integrated model, we demonstrate how the combined use of these methods leads to more precise estimations and improved interpretability compared to using either method in isolation.

For a concise summary of the applicable methods under different conditions of intervention knowledge and interaction complexity, we refer to Table 4. This table provides a structured overview of the methodological choices available, facilitating the selection of the most appropriate approach based on the specific characteristics of the problem at hand.

## Example 2

### Description

Example based in the real life case described in Rhee et al. (2017); Subbaswamy et al. (2022). In this scenario, the prevalence of sepsis (S) is the target variable. Three categories of patient data influence sepsis detection: vital signs (V) (e.g., heart rate, respiratory rate, temperature), lab test results (L) (e.g., lactate levels), and demographics (D) (e.g., age, gender). For patients who develop sepsis, physiologic data collected before the onset of sepsis is used. For non-sepsis patients, all available data until hospital discharge is considered. Time-series features, such as minimum, maximum, and median values, were derived for each variable.

Unlike vital signs, lab tests are not always ordered (O) at consistent intervals, leading to missing values. To account for this, a binary missingness indicator was introduced. However, lab test ordering practices (O) vary between hospitals, resulting in shifts in the conditional distribution P(O | S, D). These variations cause differences in missingness patterns across datasets, making it challenging to build a model that generalizes across institutions.

Standard feature selection methods fall short here because only demographics (D) and vital signs (V) remain stable after a change of institution. To address this, a Level 2 method can help build a more robust model. However, even Level 2 approaches may include spurious interactions, so refining the model further by accounting for these can lead to better results.

### Data Generating Process

$$\beta = 3.$$
$$D \sim \mathcal{N}(0,1)$$
$$S = \beta D + \mathcal{N}(0,2)$$
$$O = (5S - D + \mathcal{N}(2,1)) > 0$$
$$V = DS + \mathcal{N}(0,1)$$
$$L = S + D + 2O + \mathcal{N}(0,1)$$

$$\beta \sim \mathcal{N}(0,1)$$
$$O_{\text{int}} = (\beta SD + \mathcal{N}(2,1)) > 0$$
$$L_{\text{int}} = S + D + 2O_{\text{int}} + \mathcal{N}(0,1)$$

### Known mechanisms

$$S = f_S(D, \epsilon_S)$$
$$O = f_O(S, D, \epsilon_O)$$
$$V = f_V(D, S, \epsilon_V)$$
$$L = f_L(S, D, O, \epsilon_L)$$

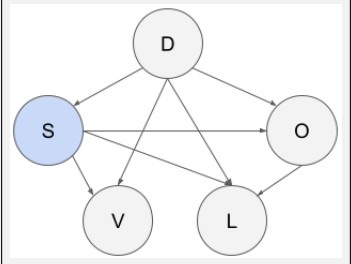

Figure 4: Causal graph for example 2.

### Results

| Model | Test MSE (SD) | Intervention MSE (SD) |
|---|---|---|
| Unconstrained | 0.653 (0.005) | 3.613 (0.060) |
| Lvl 1 - Stable Features | 2.465 (0.022) | 2.484 (0.024) |
| Lvl 2 - Interventional distribution | 0.660 (0.005) | 1.816 (0.035) |
| Lvl 2 + Interaction Constraints | 0.702 (0.005) | 1.798 (0.032) |

Table 2: Test and intervention MSE results for different models. Unconstrained is a regular predictive model using all features, Lvl 1 and 2 use the approaches described in 3 and interaction constraints uses the approach described in section 4.1 to
prevent interactions between the stable ICM.

### 3.1 Mechanism level shifts

While feature selection could address the scenario presented in Example 1, real-world cases often exhibit more complexity. A single variable can participate in multiple causal mechanisms simultaneously, making straightforward selection insufficient. This issue frequently arises in practical applications.

---

**Example 3**

**Description**

This scenario represents a causal framework for understanding how access to education (A), socioeconomic status (B), and health status (C) collectively influence career success (Y). Access to education (A) plays a pivotal role in shaping intermediate factors such as educational attainment, skills, or qualifications, which directly affect career success (Y). Socioeconomic status (B) impacts Y through access to resources like tuition or mentorship and stress levels, while also interacting with A by influencing the quality of education accessible and with C through its effect on health outcomes. Health status (C) determines the physical and mental ability to succeed in education or work. Career success (Y) emerges from a complex interplay of these factors, influenced by intermediary variables such as realized health, opportunities, and skills.

In the absence of intervention, an unrestricted model incorporating A, B, and C is optimal as it captures the full range of interactions. However, when an intervention, such as a country providing free healthcare, disrupts the natural interplay (e.g., mitigating the influence of B on C), the relationship between variables is modified. In such cases, generating a model that adjusts for these changes, rather than discarding all variables involved in the intervention, can outperform both unrestricted models and those solely reliant on causally robust features. The effectiveness of this adjusted model depends on the magnitude and nature of the intervention.

**Data Generating Process**

$\beta = 3$.
$A \sim \mathcal{N}(0, 1)$,
$B \sim \mathcal{N}(0, 1)$,
$C \sim \mathcal{N}(0, 1)$,
$Y = 3A + 2C + 3AB + 3BC + \epsilon$,
$\epsilon \sim \mathcal{N}(0, 1)$

$\beta_{int} \sim \mathcal{N}(-3, 1)$.
$Y_{int} = 3A_i + 2C_i + \beta_{int} A_i B_i +$
$\qquad + 3B_i C_i + \epsilon_i$,
$\epsilon_i \sim \mathcal{N}(0, 2)$.

**Known mechanisms**

$Y = f_1(A, \epsilon_1) + f_2(C, \epsilon_2) +$

$f_3(A, B, \epsilon_3) + f_4(B, C, \epsilon_4)$

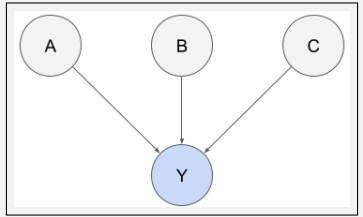

Figure 5: Causal graph for example 3.

**Results**

| Model | Test MSE (SD) | Intervention MSE (SD) |
|---|---|---|
| Unconstrained | 11.267 (4.501) | 30.849 (11.407) |
| Interaction constraints | 22.734 (6.686) | 26.411 (8.768) |
| Stable Features | 32.700 (8.050) | 35.663 (9.478) |

Table 3: Test and intervention MSE results for Decision Tree models with two intervention metrics. Unconstrained is a regular predictive model using all features, Interaction Constraints uses the approach described in section 4.1 to remove the interaction between A and B and Stable Features model uses only C.

---

| Scenario | Recommended Methods and Explanation |
|---|---|
| **No distribution shift** | Traditional machine learning methods suffice. No special causal considerations are needed for model training or evaluation. Standard supervised learning techniques perform well in this setting. |
| **Known distribution shift Causal graph available** | A known causal graph allows for precise handling of distribution shifts by selecting stable features or applying advanced methods such as those in Subbaswamy et al. (2022), provided the assumptions are met. Knowledge of the graph structure also enables the detection and prevention of interactions related to the known shift. |
| **Known distribution shift Causal graph unavailable** | Even without a causal graph, domain expertise about the shift often exists. This knowledge can inform the identification of stable mechanisms and guide feature selection and interaction constraints. |
| **Unknown distribution shift Causal graph available** | Methods such as Cui & Athey (2022) offer stability guarantees under certain conditions, even when the nature of the shift is unknown. Knowledge of the graph structure also enables the detection and prevention of spurious interactions. |
| **Unknown distribution shift Causal graph unavailable** | In the absence of both graph and shift knowledge, unstructured expert knowledge can help hypothesize stable variables and mechanisms. These can inform feature selection and assumptions of independence between mechanisms. |

Table 4: Method Selection Guide for Different Causal Learning Scenarios

In Example 3, we illustrate a scenario where variable selection alone fails to provide a reliable solution. In such cases, simply removing or retaining variables does not adequately capture the underlying causal structure. Instead, the functional form of the model must be carefully constrained to ensure robustness, particularly in out-of-distribution settings. This need for structural constraints is not just theoretical, it emerges in many real-world problems where interventions, dependencies, and latent factors influence observed outcomes.

To further demonstrate the relevance of this issue, we present a real-world case study in Experiment 5.3. This experiment highlights how failing to account for overlapping causal mechanisms can lead to poor generalization, and how imposing appropriate functional constraints results in a more robust model.

### 3.2 Preventing interactions between variables in nonlinear models

To preserve the independence structure implied by the Independent Causal Mechanisms (ICM) assumption, models must be appropriately constrained to prevent interactions between variables. These constraints can be formulated by requiring that the mixed partial derivatives between variables belonging to distinct mechanisms, and therefore expected not to interact, are equal to zero.

Formally, given a set of $K$ Independent Causal Mechanisms (ICMs) in the data generating process, let $G_k$ denote the subset of variables involved in the $k$-th ICM. The model $f(X)$ must adhere to the following constraints:

$$\frac{\partial^2 f}{\partial x_i \partial x_j} \not\equiv 0 \implies \exists k \in \{1, \dots, K\} \text{ such that } (i, j) \in G_k. \tag{2}$$

or by an analogous expression for categorical variables including finite differences (detailed in appendix C). Note that two variables may belong to the same ICM without necessarily interacting.

In the general case, the data generating process is composed of multiple ICM, each having its own input variables which can be shared or even be the same. For instance, in agriculture, crop yield is influenced by multiple independent causal mechanisms, such as rainfall affecting soil moisture, temperature regulating

metabolic rates, and sunlight driving photosynthesis. While these mechanisms may share inputs, temperature, for example, impacts both metabolism and photosynthesis, they operate independently. A robust model should capture this independence, recognizing that the relationship between inputs and outputs is governed by distinct processes rather than a single unified effect. Robust models must encode is the independence between ICM, not just between variables, including mechanisms that might share some of their inputs.

## 4 Adding the inductive bias

In this section, we introduce a comprehensive methodology for constructing more robust and interpretable predictive models by incorporating a novel inductive bias grounded in the Independent Causal Mechanisms (ICM) principle. By integrating this principle into model design, we aim to improve generalization, resilience to distribution shifts, and the interpretability of learned representations.

Although the inductive bias applies to a broader class of models, we implement this methodology specifically for tree-based and neural network models, as these are among the most widely used predictive modeling techniques in machine learning. It is important to note, however, that the proposed inductive bias is not limited to these architectures. In fact, in other cases—such as linear models—the integration of the bias can be achieved in a straightforward manner, given their simpler structure and analytical properties. Tree-based methods, such as random forest and gradient boosting frameworks (e.g., XGBoost (Chen & Guestrin, 2016), LightGBM (Ke et al., 2017), and CatBoost (Prokhorenkova et al., 2018)), are extensively used due to their effectiveness in structured data analysis, interpretability, and strong performance across various domains. Meanwhile, neural networks, including deep learning architectures, offer powerful feature extraction capabilities and flexibility in handling complex patterns in high-dimensional data.

Our approach leverages expert knowledge of variable relationships or causal graph structures and can be seamlessly integrated with existing modeling strategies to enhance robustness and interpretability.

### 4.1 Tree based models

The implementation of this methodology in tree-based algorithms is similar to the one described in Pros & Vitrià (2023) and is relatively straightforward, due to the inherent structure of decision trees and the specific way they model interactions between variables.

Decision trees, as utilized in random forests, incorporate stochasticity at two main stages of the tree-building process: data sampling and feature selection. Formally, for each tree $T_b$ in the ensemble $\{T_1, T_2, \ldots, T_B\}$, a bootstrap sample $\mathcal{D}_b$ is drawn from the original dataset $\mathcal{D} = \{(\mathbf{x}_i, y_i)\}_{i=1}^n$, where $\mathcal{D}_b \subset \mathcal{D}$.

Additionally, at each decision node in the tree, instead of considering the full set of input features $\mathcal{F} = \{x_1, x_2, \ldots, x_p\}$, a random subset $\mathcal{F}_m \subset \mathcal{F}$ with $|\mathcal{F}_m| = m \ll p$ is selected uniformly without replacement.

In ensemble methods such as gradient boosting, where the final prediction is obtained through a linear combination of individual trees, the risk of non-causal interactions interactions arises at the level of individual trees rather than the ensemble as a whole.

To remove the presence of spurious associations within individual trees, we impose a structural constraint on the splitting process. Specifically, once a variable $x_j \in \mathcal{F}$ is selected for a split, subsequent splits in its descendant nodes are restricted to variables belonging to the same Independent Causal Mechanism (ICM) as $x_j$.

Let $\mathcal{M} = \{\mathcal{M}_1, \mathcal{M}_2, \ldots, \mathcal{M}_K\}$ be a collection of $K$ subsets of the feature space $\mathcal{F}$ corresponding to distinct ICMs, such that $\bigcup_{k=1}^{K} \mathcal{M}_k = \mathcal{F}$. If a split is made on a variable $x_j \in \mathcal{M}_k$, then all subsequent splits along that branch must be on variables $x_l \in \mathcal{M}_k$.

This constraint ensures that the hierarchical decision-making process within each tree aligns with the underlying causal structure.

We implement the proposed methodology using the gradient boosting algorithm provided by the XGBoost Python library. One of its key advantages is its flexibility in controlling feature interactions, making it well-suited for incorporating causal inductive biases.

To enforce the ICM-based constraints, we leverage XGBoost's built-in feature interaction constraints hyper-parameter (Goyal et al., 2020). This feature allows the specification of a list of variable sets, where each set defines groups of features that are permitted to interact during tree construction. Note that this parameter constrains feature interactions at the node level rather than the tree level, so we followed the same approach as Pros & Vitrià (2023) when handling groups with shared features. By structuring these sets according to known causal mechanisms, we ensure that each variable can only interact with others within the same causal group, preventing spurious correlations from being learned across unrelated features.

### 4.2 Neural networks: masking

In order to apply the methodology on arbitrary neural networks, we build on the Graphical Conditioner proposed in Parafita & Vitrià (2022). Consider the original dataset $\mathcal{D} = \{(\mathbf{x}_i, y_i)\}_{i=1}^n$; consider the afore-mentioned set of distinct ICMs $\mathcal{M} = \{\mathcal{M}_1, \mathcal{M}_2, \ldots, \mathcal{M}_K\}$. The set of input features $\mathcal{F} = \{x_1, \ldots, x_p\}$ can encompass both discrete and continuous variables for generality; let us assume that any continuous feature is standardized.

Consider an arbitrary neural network architecture $f$:

$$f : \mathbb{R}^p \to \mathbb{R}^{K \times d} \tag{3}$$

$$\mathbf{x} = (x_1, \ldots, x_p) \to \mathbf{z} = (z_1, \ldots, z_K), \tag{4}$$

where $f$ transforms an input $\mathbf{x}$ into $K$ $d$-dimensional vectors $z_k$, one for each component, with $d \geq 1$. Employed directly, there is no restriction to the interaction between features $x_{j_1}, x_{j_2} \in \mathcal{F}$ from distinct ICMs ($\{x_{j_1}, x_{j_2}\} \not\subseteq \mathcal{M}_k, \forall k = 1..K$). Instead of imposing architectural constraints to $f$ (e.g., MADE for Normalizing Flows (Papamakarios et al., 2017)), we can instead alter $f$'s application to learn an inductive bias.

Let us consider an output subset $z_k$. Any time we require computing such a value, we should only employ the subset of features defined by $\mathcal{M}_k$, and any other variables must be ignored. Let us define the binary mask vector $m_k = (m_{k,1}, \cdots, m_{k,p})$ whose entries $m_{k,j}$ are 1 whenever $x_j \in \mathcal{M}_k$, and 0 otherwise. Then, we can employ a certain masking strategy $\alpha$ with mask $m_k$ to the input $\mathbf{x}$ (e.g., $\alpha_0(\mathbf{x}, m) := \mathbf{x} \cdot m$) to "hide" any input not pertaining to the current ICM $\mathcal{M}$; then, $z_k = f(\alpha(x, m_k))$. Note that $z_k$ does not depend on any input $\{x_j \mid j \notin \mathcal{M}\}$ since it never sees its values; therefore, no interaction with variables outside of the corresponding ICM can ever affect $z_k$.

If we replicate this procedure for all ICM $\mathcal{M}_k$, we can compute the concatenated vector $\mathbf{z} = (z_1, \cdots, z_K)$ with $K$ executions of $f$. Finally, if we add a single linear layer on top of $z$, $g : \mathbb{R}^{K \times d} \to \mathbb{R}^l$, with $l$ the dimensionality of the output $y$, note that $y$ cannot incorporate any interactions between ICMs, fulfilling our objective. It is important to mention that this masking must take place both during training, so that $f$ can learn to use only the appropriate variables for any $z_k$, as well as during inference, to maintain the test $\mathbf{x}$-values in-distribution w.r.t. the training distribution (masked).

Finally, the choice of masking strategy $\alpha$ may vary. In this work, we use a zero-mask $\alpha_0(\mathbf{x}, m) := \mathbf{x} \cdot m$, which ensures zero interaction derivatives between variables in separate ICMs (Eq. 6) and keeps masked inputs in-distribution, assuming that $\mathbf{x}$ is normalized.

## 5 Experiments

To evaluate the out-of-distribution performance of our models, we conduct experiments involving distributional shifts in subsets of the data. We assess this using three scenarios: a synthetic setting with generated data, and two real-world applications, real estate price prediction across different cities and rogue wave detection in distinct oceans. All the code for reproducing the experiments is available at

`https://anonymous.4open.science/r/Causal_Interaction_Constraints-4160`. All the experiments are run on a MacBook Pro with an Apple M2 Max chip and 32GB memory. Hyperparameters and evaluation metrics used to run the experiments can be found in the code or in the section B of the appendix.

### 5.1 Illustrative example: Planet orbits

Planet Orbits is a synthetic, illustrative dataset generated by simulating planetary motion using the Poliastro library (Rodríguez & Garrido, 2022). The dataset is constructed by retrieving the positions and velocities of all solar system planets relative to the Sun from the years 2000 to 2024. Each sample includes initial conditions, planetary positions and velocities, along with target values representing their states after a 30-day orbital propagation.

To design an out-of-distribution experiment, a subset of the data has the planet Venus removed. Venus is repositioned at the coordinates of the Sun with zero velocity and is excluded from the model performance evaluation.

We use two models to solve this experiment. First we use a neural network model with all the positions and velocities of the planets as inputs and their respective counterparts after 30 days as outputs. Then, we create a second model identical but adding the masking described in section 4.2 with the knowledge that planet orbits should not be affected by other planet orbits.

|              | Unconstrained  | Interactions Constraints |
|--------------|----------------|--------------------------|
| **Train**        | 0.6150 (0.01)  | 0.6402 (0.01)            |
| **Test**         | 1.4543 (0.01)  | 1.4767 (0.01)            |
| **Intervention** | 1.9205 (0.02)  | 1.5776 (0.02)            |

Table 5: Comparison of normalized Mean Average Error (MAE) with standard deviation in parentheses for Unconstrained vs. Interaction-Constrained Models on the Planet orbits Dataset. Lower values indicate better performance. Test represents data from the same distribution as training, while 'Intervention' refers to out-of-distribution evaluations.

In Table 5 we see that while the unmasked model obtains the best train and test results, the masked model has less out of distribution error. The error of the unconstrained model increases significantly in the out of distribution experiment, which suggests that the unconstrained model depended on spurious interactions with the orbit of the planet venus to obtain the predictions for the other planets. Moreover, the constrained model performs statistically significantly better in the out-of-distribution setting.

### 5.2 Real example: Real estate pricing - Known intervention, unknown causal graph.

The real estate pricing dataset is an open-source product featuring real estate listings from 2018 for Spain's three largest cities: Madrid (94,815 listings), Barcelona (61,486 listings), and Valencia (33,622 listings). Originally published in Rey-Blanco et al. (2024) , the dataset provides detailed information on asking prices and a variety of property characteristics, such as indoor features and building quality, enriched with official data from the Spanish cadastre. In addition to property details, the dataset includes relevant contextual information, such as proximity to urban points of interest. Offered as a documented R package, it is well-suited for use in price prediction models and other real estate market analyses.

Knowing that the distribution shift is caused by a change in the city of the real estate, expert information can be leveraged to reason about the causal mechanisms involved and detect unstable interactions. Since this is a common scenario, we expect LLMs to encode some relevant causal knowledge (Kiciman et al., 2023), which we used illustratively to identify potential unstable interactions. For example, the interaction between the price of a parking space and proximity to metro stations reflects differing levels of car dependency. In cities with well-developed public transportation systems, close access to metro infrastructure may reduce the necessity of a parking space, potentially weakening or reversing the expected positive relationship between parking space value and overall property price. In contrast, in more car-dependent cities, both variables

may exhibit a strong and complementary effect. Details about the expert knowledge used can be found in appendix A. We stress, however, that LLMs are not proposed as a reliable tool for causal discovery. Our use of LLM outputs serves only as an illustrative example to demonstrate how the ICM principle can be communicated or explored in broader contexts. Practitioners should not interpret LLM-inferred causal structures as validated or recommended practice. Rigorous methods for causal discovery remain essential, and further systematic study would be required before any such applications could be considered trustworthy.

To solve this experiment we use both, neural network and tree based algorithms. We use the unconstrained versions and the constrained using the procedure defined in sections 4.1, 4.2. Additionally, we add a third model that uses a popular approach, causal feature selection, discarding all features that do not generalize to a distinct city according to the expert knowledge. The experiment is repeated 500 times.

|  | Unconstrained | Interactions Constraints | Stable Features |
|---|---|---|---|
| **Train** | 0.121 (0.000) | 0.124 (0.000) | 0.203 (0.000) |
| **Test** | 0.143 (0.000) | 0.144 (0.000) | 0.231 (0.000) |
| **Intervention** | 0.413 (0.002) | 0.392 (0.001) | 0.454 (0.001) |

Table 6: XGBoost unconstrained, interaction constraints and stable features model normalized mean average error (MAE) results with standard deviation in parentheses. Lower values indicate better performance. Test refers to data from the same distribution as training, while 'Intervention' refers to out-of-distribution evaluations.

For tree-based models, the results presented in Table 6 indicate that introducing constraints leads to a modest trade-off in performance across different evaluation metrics. The unconstrained XGBoost model achieves the lowest training and testing errors, with a Train MAE of 0.121 and a Test MAE 0.143. However, this model also exhibits a high intervention error (Intervention MAE = 0.413), suggesting potential overfitting to spurious patterns that do not hold under distributional shifts.

Applying intervention constraints slightly increases the prediction errors (Train MAE = 0.124, Test MAE = 0.144) but improves robustness, reducing intervention error to 0.392—a statistically significant improvement over the unconstrained model. The stable feature model, which retains only features transferable across cities, shows a substantial increase in prediction error (Test MAE = 0.231), and intervention error (Intervention MAE = 0.454). These results illustrate the trade-off between predictive accuracy and model stability, particularly in contexts requiring generalization across varying urban environments.

|  | Unconstrained | Interactions Constraints | Stable Features |
|---|---|---|---|
| **Train** | 0.199 (0.001) | 0.204 (0.001) | 0.279 (0.001) |
| **Test** | 0.219 (0.001) | 0.212 (0.001) | 0.301 (0.001) |
| **Intervention** | 0.482 (0.005) | 0.457 (0.005) | 0.470 (0.001) |

Table 7: Neural network unconstrained, interaction constraints and stable features model normalized mean average error (MAE) results with standard deviation in parentheses. Lower values indicate better performance. Test represents data from the same distribution as training, while 'Intervention' refers to out-of-distribution evaluations.

The constrained model performance for neural networks, as summarized in Table 7, follows a similar pattern to the tree-based models, though with generally higher error values. The unconstrained neural network achieves the lowest training and test errors (Train MAE = 0.199, Test MAE = 0.219), but it also records the highest intervention error (Intervention MAE = 0.482), indicating reliance on features that may not generalize well across cities.

Adding intervention constraints has little impact on training and test errors (Train MAE = 0.204, Test MAE = 0.212), while decreasing the intervention error to 0.457, showing improved stability. The model trained with only stable features incurs the highest prediction error (Test MAE = 0.301) but obtains a lower

intervention error than the unconstrained model (Intervention MAE = 0.470). These results emphasize that while neural networks can achieve strong performance, their robustness to intervention is improved through constrained modeling and informed feature selection.

### 5.3  Real example: Wave modelling - Unknown intervention, known causal graph

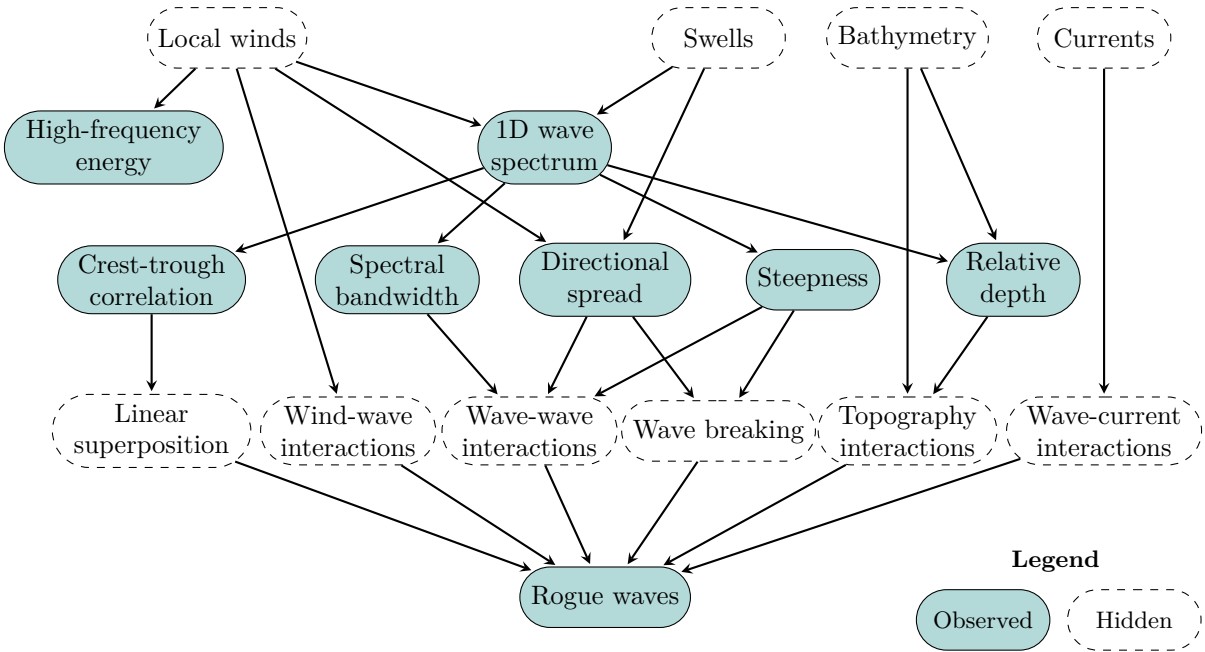

Figure 6: Causal information described in Häfner et al. (2023). The dataset consists of the variables labeled as observed, while the hidden mechanisms encode the underlying causal structure. In the original work, the authors describe all direct causes of rogue waves in the graph as independent physical effects. This goes beyond a standard causal graph by incorporating knowledge about the functional form of the rogue wave generation process. Based on the reasoning in Section 3, we can interpret this information through the lens of the Independent Causal Mechanisms (ICM) principle to identify spurious interactions. The scenario can then be addressed using the methods outlined in Section 4.2.

This dataset is introduced in Häfner et al. (2021) and is an open-source collection derived from the Free Ocean Wave Dataset (FOWD), featuring 1.4 billion wave measurements recorded by 158 CDIP wave buoys across the Pacific and Atlantic coasts of the US, Hawaii, and overseas US territories. Spanning water depths from 10 m to 4,000 m, it focuses on sea states with significant wave heights of at least 1 m. Each buoy captures sea surface elevation at 1.28 Hz, yielding over 700 years of cumulative time series data. To manage the full 1 TB dataset, an aggregated version maps each sea state to the maximum wave height of the following 100 waves, adjusting rogue wave probabilities accordingly. The final dataset includes 12.9 million data points, featuring over 100,000 extreme waves exceeding twice the significant wave height. Freely available, it is well-suited for studies in oceanography, extreme wave prediction, and maritime risk assessment. In the original work, the data is divided into distinct subsets, each with unique characteristics. These subsets are then used to assess the models' out-of-distribution error. Specifically, for each subset, a model is trained on its data and evaluated on the remaining subsets and averaged to measure generalization performance.

In the original study (Häfner et al., 2023), out-of-domain generalization is assessed while accounting for variable interactions. However, in the absence of explicit selection criteria, the authors evaluate 24 distinct models to empirically identify the one that generalizes most effectively.

In contrast, leveraging the ICM perspective allows us to incorporate expert knowledge encoded in the causal representation of the problem, as depicted in Figure 6, to systematically define valid variable interactions.

As outlined in Section 3, the causal information provides a structured framework for identifying permissible interactions by grouping variables that belong to the same ICM.

In this experiment, we evaluate four models. The first is the Unconstrained model, which uses all available variables without any restrictions. The second is E18, identified as the best-performing model among the 24 evaluated in Häfner et al. (2023). The third model, Interaction Constraints, incorporates causal information from Figure 6 to impose interactions constraints. Finally, the Causal Features model uses the same set of variables as the Interaction Constraints model but without applying the constraints, allowing us to isolate and assess the impact of the constraints themselves, independent of variable selection.

| Metric | Unconstrained | E18 | Causal Features | Interaction Constraints |
|---|---|---|---|---|
| Train | 1.04 | 0.75 | 0.80 | 0.77 |
| Test | 0.80 | 0.66 | 0.68 | 0.66 |
| Intervention | -0.63 | 0.55 | 0.44 | 0.57 |

Table 8: $L$ score results for the wave modelling experiment (higher is better). All values have been multiplied by 1000 to improve readability. Unconstrained refers to a neural network trained on all available features without constraints. E18 is the best-performing model among the 24 models tested in Häfner et al. (2023). Interaction Constraints denotes the proposed model with domain-informed constraints applied. Causal Features uses the same input features as the constrained model but without applying constraints. Due to computational time constraints, each model was run once, and standard deviations are not reported.

Table 8 presents the results of this experiment. In the test metric, all models perform comparably well, with the Unconstrained model obtaining the highest $L$ score. In the intervention metric, both Interaction Constraints and E18 exhibit comparable performance achieving higher $L$ score than the other models. This result is particularly noteworthy given the nature of the problem: strong spurious interactions that can significantly hinder generalization to out-of-distribution cases if not properly addressed. The lower performance of the both the unconstrained and the causal features model highlights this issue, demonstrating the necessity of incorporating constraints to improve robustness. In contrast to the approach of testing 24 distinct models as detailed in Häfner et al. (2023), our method leverages causal expert information from an ICM perspective to directly construct a model that achieves in-distribution and out-of-distribution performance comparable to the best among the 24 models.

# 6 Discussion and Conclusion

In this work, we have explored the critical role of spurious interactions in predictive models, particularly in the context of generalization and explainability under distribution shifts. By leveraging the Independent Causal Mechanisms (ICM) principle, we introduced a novel approach to incorporate causal knowledge into machine learning models, specifically focusing on preventing spurious interactions that can undermine model robustness. Our method, which enforces interaction constraints based on causal structures, has been demonstrated to improve out-of-distribution performance across synthetic and real-world datasets, including real estate pricing and ocean wave modeling.

Our contributions are twofold: (i) we demonstrated how interaction constraints derived from ICM, previously used mainly for causal inference, can serve as a powerful bias for predictive modeling under distribution shifts; and (ii) we developed a modified masking mechanism to integrate this bias into neural networks. Together, these advances bring machine learning models closer to the true data-generating process, leading to more reliable generalization in dynamic settings.

Our results underscore the value of integrating causal knowledge into machine learning pipelines, particularly in domains where distribution shifts are common. While traditional machine learning models excel in i.i.d. settings, their reliance on spurious correlations often leads to poor performance in real-world applications. By contrast, our causal approach provides a principled way to build models that are not only accurate but also robust and interpretable. Beyond robustness, our findings suggest that aligning model interactions with causal structures may also enhance interpretability. This alignment could help ensure that variable

contributions reflect their true causal roles, supporting transparency and trust. Investigating this further represents a promising avenue for future research.

Future work could explore extending this framework to more complex causal structures, integrating it with other machine learning techniques, and applying it to additional domains where robustness to distribution shifts is critical. Additionally, further research could investigate the trade-offs between model complexity and the incorporation of causal constraints, as well as the scalability of these methods to larger and more diverse datasets.

## Appendix A. Real estate pricing expert knowledge

As described in Kiciman et al. (2023), we query a LLM model in order to obtain causal knowledge relevant to the real estate pricing sscenario.

Queries for expert knowledge were made to the 'gpt-4' model between 2024-11-06 and 2025-01-16 using openai.ChatCompletion.create(model='gpt-4, [...]') with the following prompt:

*We need to develop a pricing model for real estate across different cities. Since the data varies by location, we are unsure which variable interactions to include or exclude. From the list provided, can you identify the top 5 pairs of variables whose interactions we should avoid adding, because their effect on price is highly dependent on the city? To clarify: we are not asking which individual variables are more city-dependent , we are asking which interactions between two variables are more influenced by the city.*

The prompt is completed with the list of variables and their descriptions from Rey-Blanco et al. (2024) and is available in the code respository.

The following interactions are obtained as potentially unstable:

- **Distance to city center - Constructed area**
  This interaction term accounts for the varying relationship between property size and distance from the city center. In cities characterized by high urban density, smaller properties near the center may command higher prices due to convenience and accessibility. Conversely, in more suburban or sprawling urban environments, larger properties located farther from the center may be more desirable. This relationship is expected to fluctuate across cities depending on local preferences and urban form.

- **Parking space price - Distance to metro**
  The interaction between the price of a parking space and proximity to metro stations reflects differing levels of car dependency. In cities with well-developed public transportation systems, close access to metro infrastructure may reduce the necessity of a parking space, potentially weakening or reversing the expected positive relationship between parking space value and overall property price. In contrast, in more car-dependent cities, both variables may exhibit a strong and complementary effect.

- **Has garden - Distance to city center**
  This term captures how the desirability of a garden varies with proximity to the city center. In some urban areas, gardens are scarce and therefore highly valued in central locations. In others, gardens are more prevalent in peripheral areas, and their marginal contribution to property value may decline with increasing distance from the city core.

- **Has south orientation × Latitude**
  The value of southern exposure, typically associated with increased sunlight, may differ according to the city's latitude. In cities located at higher latitudes, where sunlight is more limited, the orientation of a property becomes more significant. In contrast, in cities near the equator, this feature may play a less important role in determining property value.

- **Has lift × Floor level**
  The presence of an elevator interacts with the floor level of a property, with the magnitude and direction of this interaction varying by urban context. In cities with numerous high-rise residential buildings, elevators are critical and substantially influence property value, particularly for higher-floor units. However, in cities with predominantly low-rise housing, this interaction may be less pronounced.

After obtaining the list of unstable interactions, we prompt again the LLM to discover the variables that are in the same ICM as the ones involved in the interactions. The prompt used is:

*Which of the following variables are directly related to the variable [variable name]?*

Again, the prompt is completed with the list of variables and their descriptions from Rey-Blanco et al. (2024).

After removing the unstable interactions previously discovered, we obtain the following groups which are used for the implementation of the methods in Experiment 5.2.

1. DISTANCE_TO_CITY_CENTER, DISTANCE_TO_MAIN_STREET, FLOORCLEAN

2. CONSTRUCTEDAREA, ROOMNUMBER, BATHNUMBER, HASPARKINGSPACE, HASSWIMMINGPOOL, HASGARDEN

3. PARKINGSPACEPRICE, HASPARKINGSPACE, ISPARKINGSPACEINCLUDEDINPRICE, DISTANCE_TO_CITY_CENTER, HASLIFT

4. DISTANCE_TO_METRO, DISTANCE_TO_MAIN_STREET, DISTANCE_TO_CITY_CENTER, LATITUDE, LONGITUDE

5. HASGARDEN, CONSTRUCTEDAREA, HASSWIMMINGPOOL

6. HASSOUTHORIENTATION, ROOMNUMBER, CONSTRUCTEDAREA, HASGARDEN

7. LATITUDE, DISTANCE_TO_CITY_CENTER, CONSTRUCTEDAREA

8. HASLIFT, CONSTRUCTEDAREA, PARKINGSPACEPRICE, DISTANCE_TO_CITY_CENTER

9. FLOORCLEAN, CONSTRUCTEDAREA, DISTANCE_TO_CITY_CENTER

## Appendix B. Hyperparameters and evaluation metrics.

In this section, we detail the hyperparameters and evaluation metrics used in the three experiments conducted. Each experiment involved training machine learning models under varying configurations.

**Planet orbits**

**Neural Network:**

- Optimizer: Adam

- Learning Rate: 0.0001

- Number of Hidden Layers: 2

- Neurons per Hidden Layer: (32, 12)

- Number of Training Epochs: 100

- Batch size: 16

**Evaluation metric**

Mean Average Error:

$$\text{MAE} = \frac{1}{n} \sum_{i=1}^{n} |y_i - \hat{y}_i|$$

**Real estate pricing**

**Neural Network:**

- Optimizer: Adam

- Learning Rate: 0.0006

- Number of Hidden Layers: 2

- Neurons per Hidden Layer: (80, 20)

- Number of Training Epochs: 50

- batch size: 512

**XGBoost:**

- Default Python XGBOOST hyperparameters with Number of Boosting Rounds = 100.

**Evaluation metric**

Mean Average Error:

$$\text{MAE} = \frac{1}{n} \sum_{i=1}^{n} |y_i - \hat{y}_i|$$

**Wave modelling**

**Neural Network:**

- Optimizer: Adam

- Learning Rate: 0.0001

- Number of Hidden Layers: 3

- Neurons per Hidden Layer: (64, 32, 16)

- Number of Training Epochs: 50

- Batch size: 1024

**Evaluation metric**

For the rogue wave modeling experiment we use $L$ score as defined in Häfner et al. (2023), the log of the likelihood ratio between the predictions of the model and a baseline model that predicts the empirical base rate $\overline{y}_k = \frac{1}{n} \sum_{i=1}^{n} y_k$, averaged over all environments $k$:

$$L(p, \overline{y}) = \frac{1}{n} \sum_{k=1}^{n} (I(p_k) - I(\overline{y}_k))$$

$$I(x) = x \cdot \log(x) + (1 - x) \cdot \log(1 - x)$$

## Appendix C. Discrete and Mixed variables constraints expression.

Given a set of $K$ Independent Causal Mechanisms (ICMs) in the data generating process, let $G_k$ denote the subset of variables involved in the $k$-th ICM. The model $f(X)$ must adhere to the following constraints: if the interaction between variables $x_i$ and $x_j$ is present, then the pair belongs to some mechanism.

Formally, if $f$ is such that

$$\begin{cases} \dfrac{\partial^2 f}{\partial x_i \, \partial x_j} \not\equiv 0 & \text{if both } x_i, x_j \text{ continuous,} \\[2ex] \exists \, u \neq v \in \mathcal{L}_i, \, w \neq z \in \mathcal{L}_j \, : \, \Delta_{i,u,v} \Delta_{j,w,z} f \neq 0 & \text{if both } x_i, x_j \text{ categorical,} \\[2ex] \exists \, w \neq z \in \mathcal{L}_j \, : \, \Delta_{j,w,z} \dfrac{\partial f}{\partial x_i} \not\equiv 0 & \text{if } x_i \text{ continuous, } x_j \text{ categorical,} \end{cases}$$

then,

$$\exists \, k \in \{1, \ldots, K\} \text{ such that } (i, j) \in G_k,$$

where $\mathcal{L}_l$ denotes the set of levels for categorical variable $x_l$, and the finite difference operator is defined as $\Delta_{l,a,b} g(\mathbf{x}) = g(\mathbf{x}_{l=b}) - g(\mathbf{x}_{l=a})$ with $\mathbf{x}_{l=c}$ indicating the vector $\mathbf{x}$ with its $l$-th component set to level $c$ (holding all other variables fixed).

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
