# OpenReview forum: "Improving Generalization in ML models via Causal Interaction Constraints"
_TMLR — Rejected by TMLR_

### Review · Reviewer_CnxN · 2025-07-31

**Summary Of Contributions:**

The paper proposes to constrain interactions between variables that are not part of the same causal mechanism, in order to improve out-of-domain performance and interpretability. The method is implemented for decision tree-based models and neural networks. Experiments on a variety of synthetic and real-world data sets demonstrate the utility of the approach.


**Strengths**

1. The paper is very easy to read. The examples are easy to follow and well-chosen. Figures are very nice. The author took great care that readers with different backgrounds can follow. The paper should be accessible to a broad audience.

2. The method seems to work well and experiments on interesting and creative data sets are conducted (specifically, rogue wave detection and removing venus from the planetary orbits simulation as OOD task is very interesting, in my opinion). I believe that the experimental evaluation is the main strength of the paper.

3. Code is available (reproducability is paramount).

**Weaknesses**

1. I am not sure if it makes sense to motivate the used method from causality (and the ICM principle, specifically). The authors show that it helps in terms of OOD performance to respect knowledge about interaction constraints, but this goes beyond causality, no? From what I understand, the paper says that variables from different causal mechanisms should not interact and that this should be enforced in the model design. However, even if $x_1$ and $x_2$ are part of the same mechanism, it could be that they do not interact and it would then make sense to enforce this, no? The question of modeling variable interactions seems much broader to me and I am not sure whether the connection to the ICM principle is that meaningful.

2. I am not sure if I understand what the contribution of the paper is. Can the paper be summarized as *"if $x_1$ and $x_2$ are not part of the same causal mechanism, it makes sense to constrain their interaction"*? (If yes, I do not understand why the paper spans 17 pages; see **weakness 3**). In terms of methodology, it seems that the authors rely on existing approaches (?), so the main contribution would then be about how the ICM principle relates to variable interactions (however, as mentioned in **weakness 1**, the meaningfulness of this relation is not really clear to me yet).

3. In my opinion, the paper is too verbose. While I do appreciate that the authors use intuitive examples and explanations, it is a bit too much for my taste. For the current version, I am afraid that it will be hard for the paper to gain traction because of how unnecessarily long it is. I would consider cutting the initial 11 pages down drastically (everything before section 3.2) and generally adopting a more direct style of writing (see **requested changes** for more concrete advice).

**Additional Comments:**

None

**Audience:**

Yes

**Audience Explanation:**

Out-of-domain generalization continues to be a problem in machine learning and the authors provide a method for tackling this.

**Claims And Evidence:**

Yes

**Claims Explanation:**

There are many experiments that convincingly bring the point across (but I believe there are some minor issues regarding correctness).

**Requested Changes:**

I will proceed to list requested changes one-by-one. Points that are critical for meeting the acceptance criteria are highlighted by *(critical)*.

**Title**

Consider inserting a newline after "ML Models" for better visual appearance.

**Abstract**

- This is a subjective matter, but I am not a fan of having multiple paragraphs within the abstract. Consider collapsing it into a single paragraph.

- *"tree-based models"* What is meant by that? A decision tree? Consider clarifying what this means.

**1 Introdution**

- *"Causal Machine Learning, which aims to enhance conventional machine learning techniques by integrating causal knowledge"* Maybe I do not understand this sentence correctly, but causal machine learning is not just about integrating causal knowledge. Many methods aim at inferring causal relationships from data (causal discovery), meaning that causal knowledge is inferred and not merely integrated. You may consider reformulating this sentence.

- What I am missing in the introduction is how this *novel approach* works on a high level. All that the reader gets to know is that it is based on ICM, but that is not enough to understand what the contribution really is. I would kindly ask the authors to give more insights into their approach.

**2.1 Causal Machine Learning**

- *Notably, Parascandolo et al. (2018) introduce a framework* I am familiar with this paper and I never understood why it is about learning causal mechanisms. To me, the paper seems to be more about decomposing a distribution into multiple sub-distributions, but it is not clear to me why these would necessarily correspond to causal mechanisms. Using a simplified setup where we have two variables $x$ and $y$ where $x$ causes $y$: Why would competition between experts guarantee that we will learn one mechanism for $p(x)$ and another one for $p(y | x)$ and not, e.g., $p(y)$ and $p(x | y)$ (which would be anti-causal)? Maybe the authors can explain this to me... I would be grateful.

**3 Interaction Selection**

- *"These spurious interactions can occur when there is a high degree of collinearity among some (or all) of the predictor variables in the training data. Since the data-generating process does not contain such interactions, predictions based on them will not be robust to interventions and thus to distribution shifts."* I think the authors need to be a bit more careful in this formulation. There may actually be a *causal interaction* between $x_1$ and $x_2$. I suggest writing *Since the data-generating process **may not** not contain such interactions, predictions based on them **may** not be robust to interventions and thus to distribution shifts*.

- *(critical)* *we can leverage the methodological approaches outlined in Section 2.3.* Section 2.3 does not exist. Please correct this.

- *"the interaction selection approach described in the previous section"* At this point of reading the paper, it is not clear what the *interaction selection approach* is. Would it be possible to clearly state what this *interaction selection approach* is? Ideally, write an algorithm box entitled *interaction selection approach* and write concrete pseudo code that takes causal domain knowledge and data as an input and outputs a predictive model. If you do this, you could perhaps resolve my **weaknesses 1. and 2**. in a single step.

- The caption of figure 3 has a typo. *"distinc" -> "distinct"*

- *(critical)* Example 1 & 2: It is great to see that this *interaction constraints method* works. However, it is not clear to me (1) what this *interaction constraints method* is and (2) how it relates to the *interaction selection approach*. Are these methods the same? If yes, choose one name for both of them and if not, how do they work together? Again, algorithm boxes would help.

**3.2 Preventing interactions between variables in nonlinear models**

- *(critical)* equation (2): This seems to be the central equation of the paper. However, I believe it is not entirely correct. Specifically, I believe that this is not an "if and only if". If two variables are in the same mechanism, then they may still not interact. Thus, the equation should read: $\frac{\partial f(X)}{\partial x_i \partial x_j} \neq 0 \implies (i, j) \in G_k, \forall k$. Also, consider showing this equation much earlier in the paper.

**Adding the inductive bias**

- Would it be possible to highlight more clearly what has already been done in prior work and which methodological contribution comes from this paper? It is generally not really clear to me.

---

> ### Author Response · Authors · 2025-08-26
> **Review answer (1)**
>
> We thank the reviewer for the careful reading and valuable feedback on our submission and hope our response could solve your concerns.
>
> ---
>
> *_Reviewer comment: (critical) we can leverage the methodological approaches outlined in Section 2.3. Section 2.3 does not exist. Please correct this._*
>
> Thanks for pointing this out, this was a typo in the manuscript. It is section 2.2; we fix it and add a brief reminder of the nature of the methods.
>
> ---
>
> *_Reviewer comment: (critical) Example 1 & 2: It is great to see that this interaction constraints method works. However, it is not clear to me (1) what this interaction constraints method is and (2) how it relates to the interaction selection approach. Are these methods the same? If yes, choose one name for both of them and if not, how do they work together? Again, algorithm boxes would help._*
>
> Thank you for pointing this out. We agree that the relation between models with interaction constraints and interaction selection was not clearly explained. In the revision we clarify by adding an algorithm box explaining how interactions are selected (interaction selection approach) and specify that the approach described in section 4.1 (Tree-based) is used to add the constraints into the models of the examples.
>
> ---
>
> *_Reviewer comment: (critical) equation (2): This seems to be the central equation of the paper. However, I believe it is not entirely correct. Specifically, I believe that this is not an "if and only if". If two variables are in the same mechanism, then they may still not interact.[...]_*
>
> Thank you for pointing this out. We agree with your observation. This issue reflects a limitation of the causal mechanism independence definition. If two variables in a system, such as $y = f(a, b)$, belong to the same mechanism but do not interact, they can be written as $y = g(a) + h(b)$, where $g(\cdot)$ and $h(\cdot)$ still exhibit mutual information (see [1]). In such cases, $g(\cdot)$ and $h(\cdot)$ are no longer causal mechanisms in the strict sense. Assessing whether these interactions should be included lies beyond the scope of this work. We change the equation and briefly reference this scenario.
>
> ---
>
> *_Reviewer comment: Consider inserting a newline after "ML Models" for better visual appearance._*
>
> We adjust the title formatting to improve its readability.
>
> ---
>
> *_Reviewer comment: This is a subjective matter, but I am not a fan of having multiple paragraphs within the abstract. Consider collapsing it into a single paragraph._*
>
> We revise the abstract into a single paragraph to enhance flow and conciseness.
>
> ---
>
> *_Reviewer comment: "tree-based models" What is meant by that? A decision tree? Consider clarifying what this means._*
>
> Thank you for pointing this out. We clarify that by "tree-based models" we refer to models such as decision trees, random forests, and gradient-boosted trees.
>
> ---
>
> *_Reviewer comment: "Causal Machine Learning, which aims to enhance conventional machine learning techniques by integrating causal knowledge" Maybe I do not understand this sentence correctly, but causal machine learning is not just about integrating causal knowledge. [...]._*
>
> We appreciate the feedback. We reformulate the sentence to reflect that causal machine learning includes both integrating causal knowledge and inferring causal relationships from data.
>
> ---
>
> *_Reviewer comment: What I am missing in the introduction is how this novel approach works on a high level. All that the reader gets to know is that it is based on ICM, but that is not enough to understand what the contribution really is. I would kindly ask the authors to give more insights into their approach._*
>
> We agree that a high-level overview would be beneficial. We have added a concise description of the methodology used along with a clarification of the contributions of this work in the introduction.
>
> ---
>
>
> *_Reviewer comment: "These spurious interactions can occur when there is a high degree of collinearity among some (or all) of the predictor variables in the training data. Since the data-generating process does not contain such interactions, predictions based on them will not be robust to interventions and thus to distribution shifts."[...]._*
>
> Thank you for noting this. We rephrase the sentence to reflect that such interactions may or may not exist in the data-generating process.
>
> ---
>
> *_Reviewer comment: "the interaction selection approach described in the previous section" At this point of reading the paper, it is not clear what the interaction selection approach is. Would it be possible to clearly state what this interaction selection approach is? Ideally, write an algorithm box entitled interaction selection approach [...]. _*
>
> We agree that additional clarity would help. We added a dedicated algorithm box describing the interaction selection approach step-by-step.
>
> ---

---

> ### Author Response · Authors · 2025-08-26
> **Review answer (2)**
>
> *_Reviewer comment: The caption of figure 3 has a typo. "distinc" → "distinct"._*
>
> We correct this typo.
>
> ---
> *_Reviewer comment: Notably, Parascandolo et al. (2018) introduce a framework I am familiar with this paper and I never understood why it is about learning causal mechanisms. To me, the paper seems to be more about decomposing a distribution into multiple sub-distributions, but it is not clear to me why these would necessarily correspond to causal mechanisms. Using a simplified setup where we have two variables x and y where x causes y: Why would competition between experts guarantee that we will learn one mechanism for p(x) and another one for p(y | x) and not, e.g., p(y) and p(x | y) (which would be anti-causal)? Maybe the authors can explain this to me... I would be grateful._*
>
> We share the same starting point as Parascandolo et al. (2018) in adopting the Independent Causal Mechanisms (ICM) assumption, though our work does not employ their unsupervised mechanism discovery method. To address the reviewer’s query regarding the bivariate case ($x \to y$): the competition between experts in their framework favors learning the causal factorization $p(x, y) = p(x)p(y|x)$ over the anti-causal $p(y)p(x|y)$ due to the ICM assumption. Under ICM, causal mechanisms ($p(x)$ and $p(y|x)$) are independent, meaning their parameters do not inform each other and remain invariant under interventions. In contrast, $p(y)$ and $p(x|y)$ are dependent, as the parameters of $p(y)$ depend on those of $p(x|y)$. The competitive training thus selects the causal factorization because it aligns with this independence. To illustrate, consider a linear example:
> $x \to y$
> $x \sim \mathcal{N}(0, \sigma_x^2)$
> $y = ax + \epsilon$, $\epsilon \sim \mathcal{N}(0, \sigma_\epsilon^2)$
> $p(x) = \mathcal{N}(0, \sigma_x^2)$
> $p(y) = \mathcal{N}(0, a^2\sigma_x^2 + \sigma_\epsilon^2)$
> $p(y|x) = \mathcal{N}(ax, \sigma_\epsilon^2)$
> $p(x|y) = \mathcal{N}\left( \frac{a \sigma_x^2}{a^2\sigma_x^2 + \sigma_\epsilon^2} y, \frac{\sigma_x^2 \sigma_\epsilon^2}{a^2\sigma_x^2 + \sigma_\epsilon^2} \right)$
> In this example, the parameter $a$ appears in both $p(y)$ and $p(x|y)$, creating mutual dependence (e.g., changing $a$ affects both distributions’ parameters). In contrast, $p(x)$’s parameters ($\sigma_x^2$) are independent of $p(y|x)$’s parameters ($a$, $\sigma_\epsilon^2$). We hope this clarifies why the framework selects causal mechanisms.
>
> ---
>
>
> *_Reviewer comment: Also, consider showing this equation much earlier in the paper._*
>
> We thank the reviewer for this thoughtful suggestion. Introducing the equation earlier is indeed an interesting option, but in our manuscript we chose to present it as the concluding point of Section 3, after providing the necessary conceptual and methodological context, to preserve the logical flow and avoid reducing clarity for readers unfamiliar with the framework.
>
> ---
>
> *_Reviewer comment: Would it be possible to highlight more clearly what has already been done in prior work and which methodological contribution comes from this paper? It is generally not really clear to me._*
>
> We revised the introduction and conclusion to more explicitly state the central goal and contributions of the paper. We explicitly separate the contributions of prior work from our own methodological innovations. The novelty lies in (i) establishing the relevance of the ICM assumption as a principled foundation for formulating inductive biases in machine learning systems designed for real-world deployment, and (ii) proposing a novel methodology that generalizes the masking strategy of [2] to systematically incorporate this inductive bias into neural network architectures.
> Despite its theoretical importance, the ICM principle has not been widely adopted or systematically considered within the practice of machine learning. This paper seeks to address this gap by introducing the principle to a broader audience of researchers and practitioners, emphasizing its relevance as a foundation for formulating inductive biases. In doing so, we aim to demonstrate that even a conceptually simple bias such as ICM can provide substantial value for the design and deployment of machine learning systems, and we advocate for its more extensive and deliberate integration into both methodological research and applied settings.
>
> ---
>
> **References**
> [1] Giambattista Parascandolo, Niki Kilbertus, Mateo Rojas-Carulla, and Bernhard Schölkopf. Learning independent causal mechanisms. In International Conference on Machine Learning, pp. 4036–4044. PMLR, 2018.
> [2] Álvaro Parafita and Jordi Vitrià. Estimand-agnostic causal query estimation with deep causal graphs. IEEE Access, 10:71370–71386, 2022.

---

> ### Comment · Reviewer_CnxN · 2025-08-27
>
> I thank the authors for the revision. Most of my points were addressed and I believe the manuscript has greatly improved. Although this is a subjective point and not decisive for acceptance, I still believe the paper is way too verbose in relation to the effective content that it provides. For future submissions, I recommend the authors to be more direct and concise.
>
> I also have two remarks:
>
> 1. I found a new typo in the revised introduction section: *"from Parafita & Vitrià (2022)3 that"*. I think the "3" should not be there.
>
> 2. Regarding Parascandolo et al. (2018), I am not convinced of the author's response. I understand what the ICM principle is and the example is correct. However, the example does not imply that these mechanisms can be identified from observational data. In fact, it is well-known that independent mechanisms cannot be identified from observational data, in general (see Section 7 and specifically Proposition 7.1 of [1]). I do not believe that Parascandolo et al. (2018) have access to interventional data or causal knowledge or make assumptions like additive noise etc. (please correct me if I am wrong). I know that the paper is often cited for learning causal mechanisms, but I am not convinced that this is justified. If indeed it is a common misconception, we should actively avoid reinforcing it further. Hence, the authors may either 1) convince me that this work is related to learning causal mechanisms; or 2) consider removing the reference.
>
> [1] Peters, Jonas, Dominik Janzing, and Bernhard Schölkopf. Elements of causal inference: foundations and learning algorithms. The MIT press, 2017.

---

> > ### Author Response · Authors · 2025-08-28
> >
> > We thank the reviewer for the helpful comments. The noted typo has been corrected.
> >
> > We would also like to emphasize once more  that our approach does not employ the unsupervised discovery method presented in [1]. While [1] introduces certain assumptions about the data generation process, we agree with the reviewer that the connection between the assumptions and the identifiability of causal mechanisms is not explicitly addressed.
> >
> > In light of these concerns, we have revised our manuscript to instead refer to [2], where the assumptions are stated more clearly, as an illustrative example of an invariant mechanism learning strategy.
> >
> >
> > [1] Giambattista Parascandolo, Niki Kilbertus, Mateo Rojas-Carulla, and Bernhard Schölkopf. Learning independent causal mechanisms. In International Conference on Machine Learning, pp. 4036–4044. PMLR, 2018.
> >
> > [2] Zhongqi Yue, Qianru Sun, Xian-Sheng Hua, and Hanwang Zhang. Transporting causal mechanisms for unsupervised domain adaptation. In Proceedings of the IEEE/CVF International Conference on Computer Vision, pp. 8599–8608, 2021.

---

> > > ### Comment · Reviewer_CnxN · 2025-08-28
> > >
> > > I thank the authors for the response and for the revision. I checked the alternative reference and it seems more clearly related to causality than Parascandolo et al. (2018), where I have great doubts whether it has something to do with causality at all (in spite of the paper's title).

---

### Review · Reviewer_iCME · 2025-08-12

**Summary Of Contributions:**

Authors proposed a methodology to improve the generalisation performance of predictive models by adding causal constraints.

- Weaknesses
    - Lack of comparison with state of the art: authors propose to add some constraints between variables to improve the predictive performance but do not compare their proposed approach with other existing methods (structural causal models and causal Bayesian networks).
    - Unclear methodology reference: which causal framework is used? Are the causal models given or learnt from data? How do you deal with unobserved variables?

**Audience:**

Yes

**Audience Explanation:**

The topic is interesting due to the rapid development of the causal machine learning field.

**Claims And Evidence:**

Yes

**Claims Explanation:**

Including prior knowledge about the data generating mechanism is a valuable option to make existing methods more robust, still more work is needed to make this intuition mature. More in the requested change section.

**Requested Changes:**

- Improve the "related works" section 2: the most updated references are three years old, and many of them are older.
- Explain clearly whether the causal model is given on learnt from data. In some part of the paper this is not clear:
    - Section 2.1 the "important features/mechanisms" are learnt,
    - Section 3 the interactions are set as per Equation 1,
    - Section 3 the examples are cases in which the mechanism is given.
- Why are SHAP values relevant to your approach? What are the confidence intervals for Figure 3?
- What do you mean by "Causal Learning" in Table 2?
- Explain clearly why Section 4 takes into consideration only trees and neural networks. What is the rationale for such cherry picking?
- Depict the causal models for each experiment. It is quite hard to understand exactly why such experiments should be relevant for the proposed task.
- Since your aim is to add some "causal constraints" to predictive models you should compare these predictions against actual causal models, such as structural causal models and causal Bayesian networks to evaluate the improvement w.r.t. your approach.
- How do you deal with unobserved variables in Figure 6?

---

> ### Comment · Reviewer_CnxN · 2025-08-15
>
> I am another reviewer. Thank you for writing the review. I will comment on two points:
>
> > Lack of comparison with state of the art: authors propose to add some constraints between variables to improve the predictive performance but do not compare their proposed approach with other existing methods (structural causal models and causal Bayesian networks).
>
> The authors write clearly that they only require knowing a subset of the structural relationships. Thus, *comparing to structural causal mechanisms* does not make sense, since their method is directly based on knowing (or estimating) some parts of the structural causal model (SCM), see for example 1. By the way, every SCM entails a causal Bayesian network (but not vice versa, see Proposition 6.3 of [1]), so that *comparison* does also not really make sense in my opinion.
>
> > Unclear methodology reference: which causal framework is used? Are the causal models given or learnt from data?
>
> I believe that this is all quite clear. The authors use the structural causal model framework (page 5 and example 1) and some relationships of the SCM are given, not learned (again, example 1 tells that very clearly). Only the structural equations of the SCM are learned.
>
> [1] Peters, Jonas, Dominik Janzing, and Bernhard Schölkopf. Elements of causal inference: foundations and learning algorithms. The MIT press, 2017.

---

> ### Author Response · Authors · 2025-08-26
> **Review answer**
>
> We thank the reviewer for the careful reading and valuable feedback on our submission. We address the requested changes point by point and clarify our contributions accordingly.
>
> ---
>
> *Improve the "related works" section. The most updated references are three years old, and many of them are older.*
> Thank you for pointing this out. We update the related works section with recent literature to ensure our discussion reflects the current state of the field.
>
> ---
>
> *Clarify whether the causal model is given or learnt from data. In some part of the paper this is not clear: in one section the "important features/mechanisms" are learnt, in another the interactions are set as per Equation 1, and in the examples the mechanism is given.*
> Thank you for pointing this out. In our approach, the mechanisms are assumed to be given or obtained through domain expertise, empirical studies, automated tools such as LLMs, etc. The discussion of related methods where mechanisms may be learnt is not part of our proposal. We have revised the relevant section to explicitly clarify this distinction.
>
> ---
>
> *Why are SHAP values relevant to your approach? What are the confidence intervals for Figure 3?*
> To assess the extent to which a predictive model captures the underlying causal structure, we consider the correlation between the SHAP values of the predictive model and those derived from the true causal model. SHAP values provide a principled attribution of the relative contribution of each input variable to model predictions, thereby offering an interpretable representation of the model’s reliance on different features. By computing the correlation between the attribution profiles of the predictive and causal models, we obtain a quantitative measure of behavioral similarity. A high correlation indicates that the predictive model assigns importance to variables in a manner consistent with the true causal mechanisms, whereas a low correlation suggests reliance on spurious or non-causal associations.
> We update the plot and add the standard deviation in the caption.
>
> ---
>
> *What do you mean by "Causal Learning" in Table 2?*
> Thanks for pointing this out, we agree this phrasing is ambiguous. We update to Causal Machine Learning.
>
> ---
>
> *Explain clearly why only trees and neural networks are considered. What is the rationale for such cherry picking?*
> In this work, we restrict our analysis to decision trees and neural networks. This choice is motivated by two main considerations: (i) decision trees and neural networks are widely recognized as representative classes of interpretable and flexible models, respectively, thereby covering two contrasting yet prominent paradigms in machine learning, and (ii) both model families are sufficiently tractable to incorporate the proposed causal constraints in a principled manner. It is important to note, however, that the proposed inductive bias is not limited to these architectures. In fact, in other cases—such as linear models—the integration of the bias can be achieved in a straightforward manner, given their simpler structure and analytical properties. We will clarify this rationale and acknowledge the limitation, suggesting extension to other model classes as future work.
>
> ---
>
> *Depict the causal models for each experiment. It is quite hard to understand exactly why such experiments should be relevant for the proposed task.*
> Each experiment represents an out-of-domain scenario, with training and test distribution differences detailed. The first experiment is a simple illustrative experiment for intuition; the second is a real scenario with unknown causal structure but known intervention; and the third is a real scenario with known causal structure but unknown intervention.
>
> ---
>
> *Since your aim is to add some "causal constraints" to predictive models, you should compare these predictions against actual causal models, such as structural causal models and causal Bayesian networks to evaluate the improvement w.r.t. your approach.*
> We thank the reviewer for this suggestion. Structural causal models (SCMs) and causal Bayesian networks require stronger assumptions and detailed knowledge of the data-generating process, which may not be available in some practical settings. Our approach, in contrast, focuses on incorporating causal constraints into predictive models without requiring full knowledge of the underlying causal graph. Because of these fundamental differences, a direct quantitative comparison is not straightforward.
>
> ---
>
> *How do you deal with unobserved variables in Figure 6?*
> Following [1], unobserved variables are not included in the data and thus not directly used in the model. However, they are accounted for when defining valid interactions based on the causal structure.
>
> ---
> **References**
> [1] Dion Häfner, Johannes Gemmrich, and Markus Jochum. Machine-guided discovery of a real-world rogue wave model. Proceedings of the National Academy of Sciences 120(48):e2306275120,2023.

---

### Review · Reviewer_6jGj · 2025-08-15

**Summary Of Contributions:**

# Summary

This paper advocates for increasing robustness in ML models by leveraging the ICM viewpoint. In particular, it proposes explicitly respecting modularity of a (causal) generative process by enforcing a second derivative condition in predictive estimates. Variants of this for tree-based models and neural networks are presented, and tested on a few simple experiments.

# Strengths and Weaknesses

### Strengths

The paper is well-written and clear. It makes coherent arguments for causal inference and ICM-like constraints in machine learning models. The examples are overall clear and helpful to the reader as well.

### Weaknesses

**Novelty**

It's not particularly clear to me what is novel in this submission. The majority of the paper is written in somewhat of a tutorial style, and besides the interaction constraint, it is not clear what is novel (or claimed as novel).

Regarding the claimed novelty of the interactions, this seems quite similar to the (uncited, though very recent) publication [1]. While [1] is focused on treatment effect estimation, I think the "Spurious Interactions Principle" is quite general and intuitive. Implementing a masking mechanism such as proposed in this paper is listed as future work, but I do not think it is novel enough to rise to the level of "interesting to the TMLR audience." Similarly, the tree-based implementation mostly leverages the existing interaction constraints in XGBoost, and it is unclear how much this differs from [2].


**Experimental Validation**

I also question the value of the experiments, in the sense that they seem fairly straightforward from a causal perspective. In two of the examples, the causal graph is known, in which case encoding causal constraints is straightforward and has unsurprising benefit.

In the third case, LLMs are used to determine ICMs; the use of LLMs in causal discovery (or causal discovery-like tasks) has already appeared several times in the literature [3, 4], with corresponding critiques and reviews [5]. The consistent problem is it is unclear when LLMs will or will not work when not augmented by some other CD method on top. I also don't think this example is "hard" enough, in the sense that the average person could likely come up with a similar grouping within a few minutes. In other words, I am critical of the extent in which the causal graph is actually "unknown" here, and I don't think the experiment properly ablates for what might happen under "bad" groupings in a more difficult experiment (where the average ML engineer may not have a good a priori idea of what the ICMs are).

**The Interaction Criterion**

Moreover, the interaction criterion is not obviously correct to me. By equation 2, the criterion is given as

$$
\frac{\partial^2 f}{\partial x_i \partial x_j} \neq 0 \iff (i, j) \in G_k, \forall k.
$$

First, why would we require that the two variables are in all mechanisms, rather than any mechanism? Shouldn't a single mechanism be enough to allow interaction?

As a secondary point, what is the meaning of equality here? Is this a functional equality, in expectation, etc.? And how categorical (and especially mixed) variables are handled should be specified formally if this is central to the paper.

Finally, it's not clear to me that this should be a bidirectional implication: it seems perfectly fine for two variables within an ICM to have only linear interaction in some regions of variable space. Concretely, there may be "diminishing effects" of some sort of joint interactions, e.g., the effect on apartment price of having an AC unit after the average temperature falls below some threshold. This is especially true for categorical variables. I also don't think this is actually enforced computationally, either (only the contrapositive of the forward direction).

**Additional Comments:**

## References

[1] Pros, R., & Vitrià, J. (2025). Preventing Spurious Interactions: A New Inductive Bias for Accurate Treatment Effect Estimation. IEEE Access.

[2] Pros, R., & Vitrià, J. (2023). Exploiting Causal Knowledge During CATE Estimation Using Tree Based Metalearners. In Joint European Conference on Machine Learning and Knowledge Discovery in Databases (pp. 261-276). Cham: Springer Nature Switzerland.

[3] Kampani, S., Hidary, D., van der Poel, C., Ganahl, M., & Miao, B. (2024). LLM-initialized differentiable causal discovery. arXiv preprint arXiv:2410.21141.

[4] Verma, P., Arbour, D., Choudhary, S., Chopra, H., Solin, A., & Sinha, A. R. (2025). Think Global, Act Local: Bayesian Causal Discovery with Language Models in Sequential Data. arXiv preprint arXiv:2506.16234.

[5] Wu, X., Yu, K., Wu, J., & Tan, K. C. (2025). LLM Cannot Discover Causality, and Should Be Restricted to Non-Decisional Support in Causal Discovery. arXiv preprint arXiv:2506.00844.

[6] Zheng, X., Aragam, B., Ravikumar, P. K., & Xing, E. P. (2018). Dags with no tears: Continuous optimization for structure learning. Advances in neural information processing systems, 31.

[7] Nastl, V., & Hardt, M. (2024). Do causal predictors generalize better to new domains?. Advances in Neural Information Processing Systems, 37, 31202-31315.

**Audience:**

No

**Audience Explanation:**

Without clarification about what is novel compared to previous work (especially [1] and [2]), I do not believe this paper is of sufficient interest.

**Broader Impact Concerns:**

This paper claims to make ML models more robust, but provides no theoretical guarantees or systematic ablations when causal mechanisms are mispecified. Of particular concern to me is the LLM example, where an LLM is trusted to identify causal mechanisms. This introduces potential broader impact concerns where practitioners may use weakly hypothesized or incorrect LLM-generated causal mechanisms under the impression this increases generalization, even though this is not theoretically or empirically verified.

**Claims And Evidence:**

No

**Claims Explanation:**

As detailed above, I am not convinced that the claims are supported by the experimental evidence. It is claimed that "this leads to models that are more robust to distributional shifts and better aligned with real-world scenarios," but due to the simplicity of the included experiments, it is difficult to make such a broad claim. There are many different ways distribution shifts can arise, and I don't believe this paper systematically addresses this. There are also many ways to exploit causal knowledge (the simplest being in the structure of the model), when available, and these aren't compared to in the experiments. The paper also doesn't consider how incorrect causal mechanisms may impact performance, especially relevant due to the inclusion of LLM-generated mechanisms.

Moreover, as above, I am not sure what the novelty claims are, so it is difficult to judge the evidence of claims related to novelty.

Finally, as above, I don't think the interaction constraint is well-argued, e.g. w.r.t. bidrectionality.

**Requested Changes:**

### Critical Changes

- Clarify what the goal of the paper is, and what parts are claimed to be novel.
- Include some further experiments in more difficult environments. Perhaps the environments of [7] are a good fit, as this paper showed conventional causal constraints are not super effective at improving generalization in conventional domain shift datasets.
- Please clarify and verify that the bidirectional implication of the second derivative condition is correct, and how equality is defined.

### Typos

- Page 6, please remove the "*" in $P (A, Y, B) = P (A) * P (Y |A) * P (B |Y )$.
- Page 15, "Test MAE = 0,219" should use a decimal point instead of a comma.
- Things like "table" in "Table 3" and "table 5" are inconsistently capitalized.

---

> ### Comment · Reviewer_CnxN · 2025-08-15
>
> I am another reviewer. I agree that the novelty of the paper is not clear (anyway, the novelty is not great). However, given that you write *"I'm not sure it is novel enough to rise to the level of "interesting to the TMLR audience.""* you should assume that it is interesting to the TMLR audience, according to their official statement (https://jmlr.org/tmlr/acceptance-criteria.html):
>
> > Generally, a reviewer that is unsure as to whether a submission satisfies this criterion should assume that it does.
>
> Just something to keep in mind.

---

> > ### Comment · Reviewer_6jGj · 2025-08-15
> >
> > Thanks for pointing this out, it was overly loose language on my part; I have clarified my review accordingly. I apologize for not being precise/direct enough in the original version.

---

> ### Author Response · Authors · 2025-08-26
> **Review answer (1)**
>
> We thank the reviewer for the constructive comments and careful reading of our manuscript. We address the critical changes first.
>
> ---
>
> *_Reviewer comment: Clarify what the goal of the paper is, and what parts are claimed to be novel._*
>
> We appreciate this observation. We revised the introduction and conclusion to more explicitly state the central goal and contributions of the paper. We explicitly separate the contributions of prior work from our own methodological innovations. The novelty lies in (i) establishing the relevance of the ICM assumption as a principled foundation for formulating inductive biases in machine learning systems designed for real-world deployment, and (ii) proposing a novel methodology that generalizes the masking strategy of [1] to systematically incorporate this inductive bias into neural network architectures.
> Despite its theoretical importance, the ICM principle has not been widely adopted or systematically considered within the practice of machine learning. This paper seeks to address this gap by introducing the principle to a broader audience of researchers and practitioners, emphasizing its relevance as a foundation for formulating inductive biases. In doing so, we aim to demonstrate that even a conceptually simple bias such as ICM can provide substantial value for the design and deployment of machine learning systems, and we advocate for its more extensive and deliberate integration into both methodological research and applied settings.
>
> ---
>
> *_Reviewer comment: Include some further experiments in more difficult environments. Perhaps the environments of [7] are a good fit, as this paper showed conventional causal constraints are not super effective at improving generalization in conventional domain shift datasets._*
> *_Reviewer comment: I also question the value of the experiments, in the sense that they seem fairly straightforward from a causal perspective. In two of the examples, the causal graph is known, in which case encoding causal constraints is straightforward and has unsurprising benefit._*
>
> We thank the reviewer for the suggestions regarding additional experiments and the value of our current experiments.
> Regarding the suggestion to include experiments in more difficult environments: Our approach is designed to encode known causal information as an inductive bias in predictive models, rather than to discover causal structure. For this reason, we focus on tasks where causal information is already available. Within this scope, we selected datasets that are representative and allow us to clearly illustrate the benefits of the method.
> To address the reviewer’s suggestion, we conducted a preliminary implementation of an additional experiment from [7]. In this case, causal information was not directly available. To approximate it, we followed the classification in [7], dividing features into “causal” and “other,” under the assumption that these groups correspond to distinct causal mechanisms. However, this setup does not account for the known domain shift in [7], and the amount of causal information is very limited. The results indicate that such limited information is insufficient, as both constrained and unconstrained models perform similarly, making the findings inconclusive.  Because the setup does not allow us to meaningfully demonstrate the advantages of our approach, we chose to focus the revised paper on settings where causal information is richer and the contribution of our method can be clearly explored.
>
> Regarding the comment on the value of our experiments: While the causal graphs are known in some cases, our focus is on demonstrating the benefits of encoding causal constraints as inductive bias in predictive models, not on causal discovery itself. The experiments thus serve to illustrate how encoding this information improves generalization, even in settings where the causal structure is straightforward.
> We hope these additions clarify the relevance and contribution of our experiments.
>
> ---
>
> *_Reviewer comment: Please clarify and verify that the bidirectional implication of the second derivative condition is correct, and how equality is defined._*
>
> Thank you for highlighting this important technical point. We agree with your observation. This issue reflects a limitation of the causal mechanism independence definition. If two variables in a system, such as $y = f(a, b)$, belong to the same mechanism but do not interact, they can be written as $y = g(a) + h(b)$, where $g(\cdot)$ and $h(\cdot)$ can still exhibit mutual information (see [2]). In such cases, $g(\cdot)$ and $h(\cdot)$ are no longer causal mechanisms in the strict sense. Assessing whether these interactions should be included lies beyond the scope of this work. We updated the equation accordingly and added a reference to clarify this scenario.
> The constraint is defined on the functional form of the model $f(X)$ so it is a functional equality. We update the notation to reflect that.
>
> ---

---

> > ### Author Response · Authors · 2025-08-26
> > **Review answer (2)**
> >
> > *_Reviewer comment: Page 6, please remove the "*" in …_
> > _Page 15, "Test MAE = 0,219" should use a decimal point instead of a comma._
> > _Things like "table" in "Table 3" and "table 5" are inconsistently capitalized._*
> >
> > We thank the reviewer for spotting these issues. We correct them throughout the manuscript.
> >
> > ---
> >
> > *_Reviewer comment: This paper claims to make ML models more robust, but provides no theoretical guarantees or systematic ablations when causal mechanisms are mispecified. Of particular concern to me is the LLM example, where an LLM is trusted to identify causal mechanisms. This introduces potential broader impact concerns where practitioners may use weakly hypothesized or incorrect LLM-generated causal mechanisms under the impression this increases generalization, even though this is not theoretically or empirically verified._*
> >
> > We thank the reviewer for raising this important concern. To clarify, our paper does not aim to evaluate or validate the ability of LLMs to identify causal mechanisms. Rather, we build upon prior work ([3]) that has explored this direction, and we reference it as one potential source of causal knowledge. Our contribution is orthogonal: we propose a framework for improving robustness in ML models given access to causal perspectives, regardless of how the causal mechanisms are obtained (e.g., through domain expertise, empirical studies, or automated tools such as LLMs). We agree that mis-specified causal structures, whether generated by LLMs or other means, can pose risks. While systematically studying these risks is an important direction for future research, it is beyond the scope of our current work.
> > Regarding theoretical guarantees and systematic ablation studies, we have not pursued this direction. Any bound on the difference between $P(Y | do(X), Z)$—where $X$ denotes the variables subject to out-of-domain interventions and $Z$ the remaining variables—and $P(Y | X, Z)$, corresponding to the naïve machine learning model, would inevitably be quite loose. This limitation applies in both scenarios, whether the interventions are known or unknown.
> > The looseness of the bound arises because distribution shift is highly sensitive to two factors:
> > 1. The specific mechanism affected by the intervention
> >    If the intervention alters a causal mechanism that directly influences the target variable $Y$, the resulting distribution shift can be large. Conversely, if the intervention only modifies peripheral mechanisms, the shift may be negligible. Thus, the magnitude of the change cannot be uniformly bounded without precise knowledge of which mechanism is perturbed.
> > 2. The presence of spurious interactions in the training dataset
> >    Machine learning models trained on observational data often exploit correlations that are not causally meaningful. When interventions occur, these spurious correlations tend to break down, amplifying the gap between $P(Y | do(X), Z)$ and $P(Y | X, Z)$. The extent of this amplification depends heavily on the structure and prevalence of such spurious relationships in the training data.
> > In short, the difficulty is that the impact of distribution shift is not uniform but context-dependent, varying with both the causal mechanism targeted by the intervention and the degree of spurious dependencies embedded in the dataset.
> >
> > ---
> >
> > *_Reviewer comment: First, why would we require that the two variables are in all mechanisms, rather than any mechanism? Shouldn't a single mechanism be enough to allow interaction?_
> > *_Reviewer comment: And how categorical (and especially mixed) variables are handled should be specified formally if this is central to the paper._*
> >
> > Thank you for pointing this out. Yes, a single mechanism is enough to allow interaction. We have updated the expression.
> > We thank the reviewer for this observation. We have added a reference to a new annex section where we formally specify the expressions for interaction constraints involving discrete and mixed variables. The details of how these constraints are incorporated into the algorithms are provided in Sections 4.1 and 4.2.
> >
> > ---
> >
> > **References**
> > [1] Álvaro Parafita and Jordi Vitrià. Estimand-agnostic causal query estimation with deep causal graphs. IEEE Access, 10:71370–71386, 2022.
> > [2] Giambattista Parascandolo, Niki Kilbertus, Mateo Rojas-Carulla, and Bernhard Schölkopf. Learning independent causal mechanisms. In International Conference on Machine Learning, pp. 4036–4044. PMLR, 2018.
> > [3] Emre Kiciman, Robert Ness, Amit Sharma, and Chenhao Tan. Causal reasoning and large language models: Opening a new frontier for causality. Transactions on Machine Learning Research, 2023.
> > [7] Vivian Nastl and Moritz Hardt. Do causal predictors generalize better to new domains? Advances in Neural Information Processing Systems, 37:31202–31315, 2024.

---

> > > ### Comment · Reviewer_6jGj · 2025-09-08
> > >
> > > Thank you for the detailed response. The response directly addresses some of the most pressing issues in the original submission; in particular, the revised definition of the interaction criterion seems much-improved, and I thank the authors for including the relevant definitions for discrete/mixed variables. I have a few remaining concerns, which I detail below.
> > >
> > > **1. Regarding Claimed Novelty**
> > >
> > > I thank the authors for clarifying in the introduction what the contributions of the paper are. However, the highlighted reference [1] remains uncited and undiscussed, despite introducing the same interaction criterion (at least, the initial version at submission). Similarly, [2] is cited, but the actual relationship to the proposed framework is still not discussed.
> > >
> > > **2. On the Correctness of XGBoost Interaction Constraints**
> > >
> > > I am concerned that the interaction constraints implemented in XGBoost do not actually align with the constraint suggested in the submission. The XGBoost documentation [3] cautions that the implemented constraints are at a branching level, not a path level. To be more explicit, the paper states
> > >
> > > > Let $M = \{M_1, M_2, \dots, M_K \}$ be a collection of $K$ subsets of the feature space $F$ corresponding to distinct
> > > ICMs, such that $\cup_{k=1}^{K} M_k = F$. If a split is made on a variable $x_j \in M_k$, then all subsequent splits along
> > > that branch must be on variables $x_l ∈ M_k$.
> > >
> > > But the XGBoost documentation suggests this is not actually what is enforced:
> > >
> > > > For one last example, we use [[0, 1], [1, 3, 4]] and choose feature 0 as split for the root node. At the second layer of the built tree, 1 is the only legitimate split candidate except for 0 itself, since they belong to the same constraint set. Following the grow path of our example tree below, the node at the second layer splits at feature 1. But due to the fact that 1 also belongs to second constraint set [1, 3, 4], at the third layer, we are allowed to include all features as split candidates and still comply with the interaction constraints of its ascendants.
> > >
> > > In this particular example, it seems that the initial branching is on $M_1 = \{0, 1\}$, but subsequent nodes, along the same branch, are free to branch on $M_2 = \{1, 3, 4\}$. Could the authors clarify how this does not contradict the proposed constraints?
> > >
> > > **3. Regarding Broader Impact Concerns**
> > >
> > > > To clarify, our paper does not aim to evaluate or validate the ability of LLMs to identify causal mechanisms. Rather, we build upon prior work ([3]) that has explored this direction, and we reference it as one potential source of causal knowledge. Our contribution is orthogonal: we propose a framework for improving robustness in ML models given access to causal perspectives, regardless of how the causal mechanisms are obtained (e.g., through domain expertise, empirical studies, or automated tools such as LLMs).
> > >
> > > While I understand that the use of LLMs for causal discovery is not intended as a contribution of this work, my broader impact concern remains. In particular, if the paper is positioned as "[seeking] to address [a] gap by introducing the [ICM] principle to a broader audience of researchers and practitioners," part of the target audience is practitioners -- who may have relatively little knowledge of causal machine learning -- and may construe LLM-inferred ICMs as recommended or even common practice. I understand that a systematic risk assessment may be beyond scope, but my comment was intended to highlight that the reader should be cautioned.
> > >
> > > **Minor Issues**
> > >
> > > - There are several places where `\citet{...}` is used where `\citep{...}` should be used.
> > >
> > > ## References
> > >
> > > [1] Pros, R., & Vitrià, J. (2025). Preventing Spurious Interactions: A New Inductive Bias for Accurate Treatment Effect Estimation. IEEE Access.
> > >
> > > [2] Pros, R., & Vitrià, J. (2023). Exploiting Causal Knowledge During CATE Estimation Using Tree Based Metalearners. In Joint European Conference on Machine Learning and Knowledge Discovery in Databases (pp. 261-276). Cham: Springer Nature Switzerland.
> > >
> > > [3] https://xgboost.readthedocs.io/en/stable/tutorials/feature_interaction_constraint.html

---

> > > > ### Author Response · Authors · 2025-09-10
> > > >
> > > > We thank the reviewer for the valuable feedback on our submission. We address the requested changes point by point.
> > > >
> > > > 1. Regarding Claimed Novelty
> > > >
> > > > As the reviewer correctly points out, these works introduce similar interaction criteria; however, their focus is on applications to causal inference, while our contribution lies in leveraging the criterion to study robustness under distributional shift. We have added the missing citations ([1], [2]) and clarified this distinction in the introduction section.
> > > >
> > > > 2. On the Correctness of XGBoost Interaction Constraints
> > > >
> > > > We thank the reviewer for this careful observation. We are aware of the difference between the interaction constraints as described in XGBoost’s documentation and the stricter path-level interpretation suggested in our work. Following [1], we implement a small heuristic to ensure that the intended path-level consistency is respected in practice. This guarantees the correct use of the interaction constraints parameter within our framework. We have revised the manuscript to make this point clearer.
> > > >
> > > > 3. Regarding Broader Impact Concerns
> > > >
> > > > We thank the reviewer for this important point. To address this concern, we have added a warning disclaimer for readers, clarifying that we do not propose LLM-generated knowledge as validated causal knowledge. Specifically, while we reference prior work that uses LLMs as a potential source of causal information, our framework is agnostic to the source of causal knowledge, which could include domain expertise, empirical studies, or automated tools. The disclaimer explicitly cautions practitioners and other readers that LLM-inferred ICMs are illustrative examples rather than recommended or validated practice, ensuring that our work is not misconstrued as endorsing unverified causal claims.
> > > >
> > > > Minor Issues
> > > >
> > > > We have fixed the missing \citep{...} cases.
> > > >
> > > > [1] Pros, R., & Vitrià, J. (2025). Preventing Spurious Interactions: A New Inductive Bias for Accurate Treatment Effect Estimation. IEEE Access.
> > > >
> > > > [2] Pros, R., & Vitrià, J. (2023). Exploiting Causal Knowledge During CATE Estimation Using Tree Based Metalearners. In Joint European Conference on Machine Learning and Knowledge Discovery in Databases (pp. 261-276). Cham: Springer Nature Switzerland.

---

### Decision · Action_Editor_mfkH · 2025-10-09

**Recommendation:** Reject

**Additional Comments:**

First of all, while the paper is easy to follow, the paper is written in an unconventional style. Section 2 is entirely related work, but does not have the typical structure of related work (i.e., it is not succinct, and it does not show how the paper is different from prior work in a clear enough manner). Section 3, which is supposed to be problem formulation, extends to the middle of page 12(!), and it is not precise enough.  Section 4 (approach) is easy to follow, but the contributions are rather straightforward. I am not sure how section 5 can be improved. The authors claim that additional comparisons on baselines or more challenging domains (as suggested by some of the reviewers) are not possible. This might be normal for papers regarding causality, but not the case for typical ML papers. I advise the authors to streamline their papers more in line with traditional ML papers.

In general, I suggest that the authors improve the presentation of their work, clarify the connections to prior work, and either provide more evidence for the claims or adjust the claims according to the provided evidence.

**Audience:**

No

**Audience Explanation:**

While the paper's findings are not particularly novel, it is clearly written and may be of interest to some members of the TMLR audience. However, the paper can also benefit from a clearer placement compared to prior work. Not that novelty is not a deciding factor in TMLR, so to my mind, even a good degree of overlap with prior work is considered acceptable, but it is hard to assess this from the way the paper is written. This lack of clarity is the main reason I selected no for this question.

**Claims And Evidence:**

No

**Claims Explanation:**

The claim that the approach "leads to models that are more robust to distributional shifts and better aligned with real-world scenarios" is not really justified by the present set of experiments, given that only one of the domains is interesting. The other two might not provide sufficient evidence for the claim due to their simplicity. In addition, the authors claim that "comparing to SCMs does not make sense, since their approach requires knowing some parts of the SCMs". I am fine with this, but how about the robustness baselines, which fully ignore the causality (e.g., the seminal work of Madry et al for adversarial training)?

**Resubmission Of Major Revision:**

The authors may consider submitting a major revision at a later time.